# Discovery of a terpene synthase synthesizing a nearly non-flexible eunicellane reveals the basis of flexibility

Jinfeng Li [1,2,8], Bao Chen[1,8], Zunyun Fu[3,8], Jingjing Mao[4,5,8], Lijun Liu[1], Xiaochen Chen[1], Mingyue Zheng [3], Chang-Yun Wang [2,6] ✉, Chengyuan Wang [4] ✉, Yue-Wei Guo [1,7] ✉ & Baofu Xu [1,3] ✉

Eunicellane diterpenoids, containing a typical 6,10-bicycle, are bioactive compounds widely present in marine corals, but rarely found in bacteria and plants. The intrinsic macrocycle exhibits innate structural flexibility resulting in dynamic conformational changes. However, the mechanisms controlling flexibility remain unknown. The discovery of a terpene synthase, MicA, that is responsible for the biosynthesis of a nearly non-flexible eunicellane skeleton, enable us to propose a feasible theory about the flexibility in eunicellane structures. Parallel studies of all eunicellane synthases in nature discovered to date, including 2Z-geranylgeranyl diphosphate incubations and density functional theory-based Boltzmann population computations, reveale that a *trans*-fused bicycle with a 2Z-configuration alkene restricts conformational flexibility resulting in a nearly non-flexible eunicellane skeleton. The catalytic route and the enzymatic mechanism of MicA are also elucidated by labeling experiments, density functional theory calculations, structural analysis of the artificial intelligence-based MicA model, and mutational studies.

The exploration of eunicellane diterpenoids began in 1968 when the group of *Prof*. Carl Djerassi—the Father of "the pill", isolated a diterpenoid named eunicellin from the Mediterranean gorgonian *Eunicella stricta*[1] (Supplementary Fig. 1). Eunicellin-based compounds are characterized by a distinctive 6,10-bicyclic carbon skeleton and exhibit a range of intriguing bioactivities, including cytotoxicity, anti-inflammation, antibacterial, or antifouling properties[2–4]. For example: eleutherobin[5–7], discovered from marine corals, demonstrated significant potency comparable to paclitaxel in inducing tubulin polymerization and has been subjected to clinical research (Supplementary

Fig. 1). However, corals contain a low abundance of eleutherobin and cannot be over-harvested because of their critical effect on ecology and climate[8–10], a common drawback in downstream pharmaceutical studies among natural products. Chemical synthesis has also been explored as an alternative. Nonetheless, the efficiency and yield of these synthetic routes were unfavorable[11–16]. Biosynthesis might be an excellent option for circumventing this obstacle. Nevertheless, little was known about the biosynthesis of eunicellane diterpenoids until several eunicellane terpene synthases were discovered[17–25]. The identification of Bnd4 from a soil-derived bacterium[17,19,22] marked a

[1]Shandong Laboratory of Yantai Drug Discovery, Bohai Rim Advanced Research Institute for Drug Discovery, Yantai, Shandong 264117, China. [2]Key Laboratory of Marine Drugs, The Ministry of Education of China, Institute of Evolution & Marine Biodiversity, School of Medicine and Pharmacy, Ocean University of China, Qingdao 266003, China. [3]Shanghai Institute of Materia Medica, Chinese Academy of Sciences, Zhangjiang Hi-Tech Park, Shanghai 201203, China. [4]CAS Key Laboratory of Molecular Virology and Immunology, Shanghai Institute of Immunity and Infection, Shanghai 200031, China. [5]Department of Pathogen Biology, School of Medicine and Holistic Integrative Medicine, Nanjing University of Chinese Medicine, Nanjing 210023, China. [6]Laboratory for Marine Drugs and Bioproducts, Qingdao National Laboratory for Marine Science and Technology, Qingdao 266237, China. [7]School of Medicine, Shanghai University, Shanghai 200444, China. [8]These authors contributed equally: Jinfeng Li, Bao Chen, Zunyun Fu, Jingjing Mao. ✉e-mail: changyun@ouc.edu.cn; cywang@ips.ac.cn; ywguo@simm.ac.cn; bfxu@simm.ac.cn

significant advancement in the eunicellane biosynthesis research and refuted the previously hypothesized cembrane-derived biogenesis. Subsequent AlbS, *Ec*TPS1, along with *Ba*TC-2, synthesizing different types of eunicellane skeletons, were then soon reported[18,20,21], further enriching the mechanisms of eunicellane biosynthesis (Fig. 1). But all these reported eunicellane synthases share a common yet-to-be-solved issue: the biosynthesized eunicellane scaffolds exhibit flexibility in different conformational ratios[17,26,27]. The difference is most notably observed in their nuclear magnetic resonance (NMR) spectra, which display signals that vary from sharp to broadened. Further analysis through Boltzmann distributions uncovers varying proportions[17]. The broaden signals dramatically impede NMR-based structural elucidations and bringing great challenges for chemical synthesis of eunicellane-type compounds. Previously, many researchers have mentioned the flexible characteristic of macrocycle-containing compounds. For instance, ref. 26 previously discovered that the configuration of C6 in the highly functionalized eunicellanes substantially influences conformational flipping, with (6*Z*) $\Delta^{6,7}$ being a less strained and more stable version. Similarly, ref. 28 reported that collinolactone shows two conformers in NMR spectra due to the rotating of the methyl group attached to (13*E*) $\Delta^{13,14}$. However, there has been a lack of in-depth research into why and how the structural geometries influence the conformations of these flexible compounds. To address this issue and interrogate the fascinating mechanisms involved in governing conformational changes, we sought to identify a terpene synthase responsible for synthesizing a non-flexible eunicellane scaffold. To our surprise, microeunicellols A and B[29], two bacteria-derived eunicellanes, exhibit non-flexibility even without the presence of ether bridge bonds in the ten-membered ring, a feature commonly observed in coral-derived eunicellanes that might reduce flexibility via bridging. This observation suggests that the intrinsic eunicellane scaffold of microeunicellols might be non-flexible.

In this study, the discovery and characterization of the nearly non-flexible eunicellane synthesizing terpene synthase—MicA generating the eunicellane carbon scaffold of microeunicellols is presented. Through elaborate investigations of the cryptic eunicellane synthase via isotope labeling, density functional theory (DFT) calculation, enzyme structural analysis, and substrate analogs synthesis, we reasonably postulated that the structural flexibility of eunicellane scaffolds is primarily determined by the combined stereo characteristics of the bridging carbons and the adjacent double bond.

## Results

### Discovery of the non-flexible terpene synthase, MicA

Inspired by the non-flexibility of microeunicellols A and B[29], we first proposed a biosynthetic pathway that includes a hypothesized terpene synthase, oxidase(s), and methyltransferase (Supplementary Fig. 2). Then, we targeted the producing strain and did the genome sequencing and genome mining analysis[30–34], focusing on terpene synthases. Regrettably, no evident terpene synthase gene clusters with oxidases were discovered, indicating that the obtained bacterium (Supplementary Fig. 3) was not the original producing strain, which was later confirmed by fermentation analysis. However, to the best of our luck, we found a Chinese patent detailing the production of a microeunicellol-like compound by a bacterium named *Micromonospora* sp. HM134 (Supplementary Fig. 3). Despite the inaccessibility of this patented strain, we observed that the genome of this

**Fig. 1 | Enzymatic synthesis of eunicellane scaffolds and their flexible characteristics.** Eunicellane compounds are known for their diverse and fascinating biological activities. Despite this, limited research efforts have been put into their biosynthesis. Until now, only three types of eunicellane diterpene synthases have been reported: Bnd4, AlbS, and *Ec*TPS1 (*Ba*TC-2). Bnd4, AlbS, and *Ec*TPS1 can produce eunicellane skeletons, showing varying degrees of flexibility. Unlike the

previous reports, we discovered an unusual eunicellane synthase—MicA, synthesizing an unprecedented nearly non-flexible eunicellane scaffold. Based on the elaborate functional characterization of MicA, we reasoned that the stereochemistry of the bridging carbons (C1 and C10), in conjunction with the adjacent double bond ($\Delta^{2,3}$), is pivotal in defining the structural flexibility of eunicellane scaffolds.

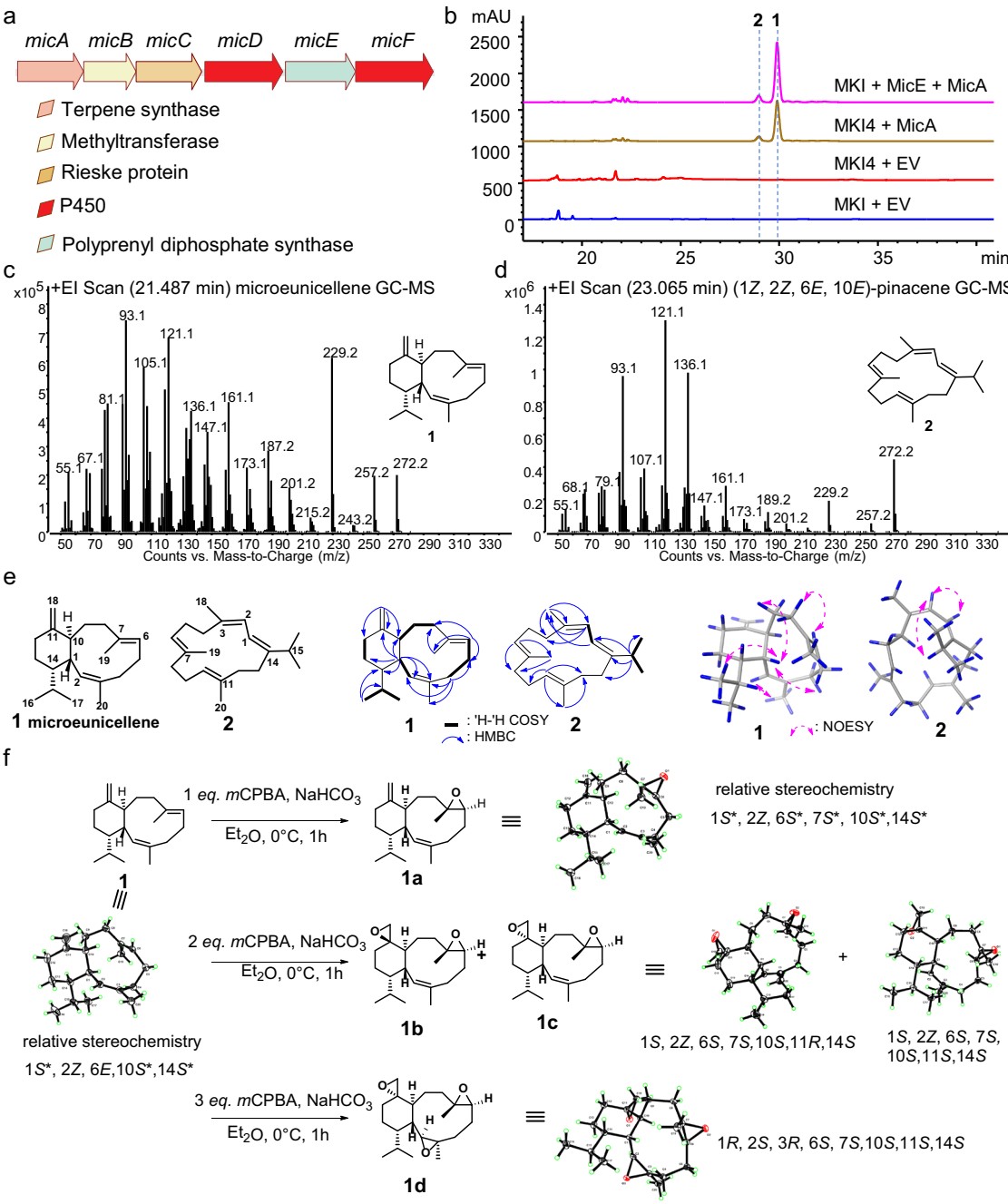

**Fig. 2 | Biochemical characterization of MicA. a** Schematic overview of microeunicellol-like biosynthetic gene cluster discovered in this study. Supplementary Fig. 3 depicts the microeunicellol-like structure seen in the patent. **b** Functional characterization of MicA utilizing previously constructed two-step kinase-based GGPP overproduction system, MKI4 (Supplementary Fig. 4)[22]. Extracted metabolites were analyzed through HPLC by monitoring at 210 nm. "EV" shown in all Figures is referred to "empty vector". GC-MS spectra of purified **1** (**c**)

and **2** (**d**). **e** Chemical structures of **1** and **2**. Critical 2D NMR information, including [1]H-[1]H COSY, HMBC, and NOESY correlations, are also shown. **f** Synthesis of epoxidized derivatives of **1** for X-ray diffraction analysis. Epoxides **1a**–**1d** were synthesized by epoxidation with *meta*-chloroperoxybenzoic acid (*m*CPBA). X-ray diffraction analysis of obtained epoxides using Cu K$\alpha$ ($\lambda = 1.5417$ Å) assigned the absolute configuration of **1**.

strain is publicly available in the NCBI database. We then easily pinpointed the microeunicellol-like gene cluster by searching for gene clusters comprising terpene synthase, oxidases, and methyltransferase (Fig. 2a). In the light of the linear operon-like gene cluster, we designated genes as *micA*–*micF* (coding protein MicA–MicF) to facilitate subsequent analysis, with MicA being the targeted terpene synthase. To assess the catalytic activity of MicA, the gene encoding MicA was commercially synthesized into the bacterial expression vector, pET28a, utilizing *BamH I* and *HindIII* as restricted cut sites while preserving the *N*-terminal His$_6$-tag.

To test our hypothesis that MicA is capable of synthesizing a non-flexible eunicellane scaffold, we first examined the activity by simultaneously expressing MicA and MKI4 (ref. 22), a two-step kinase system efficiently producing geranylgeranyl diphosphate (GGPP)[35–38], in *Escherichia coli*. This resulted in the generation of two major peaks (85% + 10%), along with several minor peaks (totally 5%) (Fig. 2b) at retention time 25-35 min, similar to previously reported diterpenes[17–19,22]. Moreover, identical peaks were observed by replacing Bnd3 in MKI4 (Fig. 2b, Supplementary Fig. 4) with *micE*, a hypothesized polyprenyl diphosphate synthase (Fig. 2a), indicating the

capability of synthesizing a diterpenoid by the targeted gene cluster. Subsequently, we purified the major products, **1** and **2**, via a large-scale fermentation (Methods) and analyzed them with GC-MS, 1D, and 2D NMR, showing consistent patterns to a new eunicellane (**1**, we named it microeunicellene) and a cembrane scaffold [**2**, (1*Z*, 2*Z*, 6*E*, 10*E*)-pinacene] (see Supplementary Information for detailed structural elucidation). The minor products are inseparable under our laboratory conditions. However, we annotated them as cembrene isomers by GC-MS analysis and comparison with the NIST database (Supplementary Figs. 5–9).

To further determine the absolute stereochemistry of **1**, which possesses three consecutive stereogenic centers (C1, C10, and C14) and two trisubstituted double bonds ($\Delta^{2,3}$ and $\Delta^{6,7}$) (Fig. 2e), **1** was subjected to crystallization. Thanks to our efficient MKI4 system, enough **1** can be easily obtained for direct or epoxidation-associated crystallization[39–41]. Fortuitously, a single crystal of **1** was observed even by directly leaving a large amount (~400 mg) in the fridge (−20 °C), consistent with its non-flexible characteristic. X-ray diffraction analysis of this single crystal of **1** (Methods and Fig. 2f) allowed us to confirm the planar structure proposed by NMR data. However, the absence of heteroatoms in **1** made determining its absolute configuration very challenging. We then introduced heteroatoms through chemical reactions, aiming to obtain new crystals. The oxidation of **1** with one to three equivalents of *meta*-chloroperoxybenzoic acid (*m*CPBA) resulted in the generation of four oxidation products: (6*S*,7*S*)-epoxy-microeunicellene (**1a**), (6*S*,7*S*,11*R*)-*bis*-epoxy-microeunicellene (**1b**), (6*S*,7*S*,11*S*)-*bis*-epoxy-microeunicellene (**1c**), and (2*S*,3*R*,6*S*,7*S*,11*S*)-*tri*-epoxy-microeunicellene (**1d**) (Methods and Fig. 2f). Fortunately, we successfully obtained single crystals of all four oxidation products using different solvent combinations (Supplementary Table 7), allowing us to determine their absolute stereochemistry through X-ray diffraction, with three of them being suitable for absolute stereochemistry determination (Fig. 2f). As a result, the absolute configuration of compound **1** was unambiguously determined to be 1*S*, 10*S*, 14*S*, 2*Z*, 6*E* (Fig. 2f). This is the first successful report of determining the absolute stereochemistry of enzyme-synthesized eunicellane diterpenes by obtaining a series of excellent single crystals.

**The effects of stereochemistry on flexibility among eunicellanes**
With MicA in hand, we are poised to pursue further investigations regarding flexibility. Benditerpe-2,6,15-triene (**6**), albireticulene (**7**), klysimplexin R (**8**), and **1** displayed a range of flexibility, from flexible to less flexible, and finally to non-flexible (Fig. 1). This pattern suggests that the combined stereochemistry likely governs the conformational changes. As such, a combined analysis of the four eunicellane synthases, Bnd4, AlbS, *Ec*TPS1, and MicA, could help clarify the underlying mechanisms. To this end, we synthesized three additional genes, expressed them in *E. coli* strains using the same MKI4 system, and successfully isolated their respective cyclization products. The NMR spectra of these eunicellane products revealed varying degrees of broadening (Fig. 3a and Supplementary Fig. 11). We meticulously analyzed the resulting structures, including the epoxidized analogs (**1a**–**1d**). The distinction between **7** and **1** lies in the configuration and positioning of the double bond. Based on these observations, we hypothesized that the stereochemistry of the bridging carbons, C1 and C10, and the adjacent double bond, $\Delta^{2,3}$, might primarily determine the flexibility of these compounds.

To verify this hypothesis, we first incubated chemically synthesized 2*Z*-GGPP (Methods and Supplementary Fig. 12) with Bnd4 and AlbS in vitro, in an attempt to obtain 2*Z*-isomers of respective eunicellane products. Regrettably, we were only able to purify compounds **9**–**12** and were unsuccessful in synthesizing our intended compounds 2*Z*-benditerpe-2,6,15-triene and 2*Z*-albireticulene (Supplementary Fig. 13). Thus, we moved to DFT calculations to substantiate our

hypothesis. We computed the Boltzmann populations of **6**–**8**, **1**, and the hypothesized 1*S*,10*R*-microeunicellene (1*S*,10*R*-**1**), 1*R*,10*S*-microeunicellene (1*R*,10*S*-**1**), 2*E*-microeunicellene (2*E*-**1**), and 1*R*,10*R*-microeunicellene (1*R*,10*R*-**1**) (Fig. 3b, Supplementary Fig. 14, Supplementary data 1) using DFT calculation at the mPW1PW91/6-31+G(d,p)//B3LYP/6-31+G(d,p) level. The results showed that the conformational space of **6** is primarily occupied by four conformations, accounting for 93.8% of the total space, approximately consistent with previous calculations[17]. Similarly, for **7**, four conformations dominate the conformational space, representing 96.9% of the total space. **8** exhibits less flexibility, with its conformational space primarily populated by two conformations. In the case of **1**, a single conformation overwhelmingly dominates the conformational space at 99.3%, exhibiting almost inflexibility (Fig. 3b). However, 1*S*,10*R*-**1** and 1*R*,10*S*-**1** exhibit flexibility when C-1 and C-10 are in a *cis* configuration, presenting two conformations (76.1% and 23.9%) for the former and three conformations (75.4%, 10.4% and 9.7%) for the latter. Similarly, 2*E*-**1** also displays flexibility when (2*Z*) $\Delta^{2,3}$ becomes (2*E*) $\Delta^{2,3}$, with two predominant conformations (80.3% and 11.2%). These findings emphasize the importance of the configurations of $\Delta^{2,3}$, C1, and C10 in determining the rigidity of the eunicellane scaffold. Specifically, the combination of a *trans*-configuration C1 and C10 with a 2*Z* double bond leads to a dominant eunicellane conformation. Additionally, the calculated results of the hypothesized 1*R*,10*R*-**1** align with those for **1**, suggesting that the *trans* configuration of C1 and C10, regardless of the absolute orientations, has the same effect on flexibility (Supplementary Fig. 14). The distinction between conformation-1 (99.3%) and conformation-2 (0.7%) of **1** resides within the hexatomic ring. Metadynamics simulations indicated that $\Delta G$ between them is ~3 kcal/mol, with a corresponding energy barrier of about 7.5 kcal/mol (Supplementary Fig. 15). Conformation-1 is capable of converting into conformation-2, yet it predominantly retains conformation-1 as its primary state. This observation is in accordance with our findings from variable temperature NMR. With the temperature rising up to 328 K, the spectrum of **1** broadens, indicating a shift in conformation and thereby revealing the presence of multiple conformations coexisting or undergoing rapid transformation (Supplementary Figs. 16 and 17). Upon detailed examination of the conformations of the remaining compounds (excluding **1**), it has been observed that their primary difference stems from the rotatable bond at C7–C19, which is attached to (6*E*) $\Delta^{6,7}$ (Fig. 3b and Supplementary Fig. 14). Therefore, we hypothesized that the combination of a *trans*-configuration C1 and C10 with a 2*Z* double bond restricts the rotation at C7–C19, leading to a dominant eunicellane conformation, which aligned with previous studies[17,26,29]. Prof. Andrei K. Yudin[42] has previously proposed that alteration in backbone stereochemistry can lead to coupled bond rotations that can result in conformational reorganizations among macrocycles. Based on that proposal, we hypothesized that the configurations of C1, C2/C3, and C10 applied similar mechanisms for adjusting the flexibility of the eunicellane macrocycle. In addition, the authors in the review[42] have provided explanations of the mechanism concerning the dihedral angles of macrocycle-containing compounds. Accordingly, we conducted a thorough conformational analysis of all related eunicellanes and discovered that compounds **1** and 1*R*,10*R*-**1** possess dihedral angles $\chi_1$ (C9-C10-C1-C2) that are greater than 130°, whereas the other compounds exhibit less than 90° (Supplementary Figs. 18 and 19). The *trans*-configuration of C1 and C10 with a 2*Z* double bond might collectively influence the rotation of the C1–C2, C1–C10, and C9–C10 bonds, resulting in augmenting the dihedral angle, thereby rendering the rotations of remaining bonds, ultimately diminishing the structural flexibility. Our observation and hypothesized mechanism align with several previous reports. For example, ref. 43 previously reported that inducing *α*-helicity through side-chain cross-linking is a strategy that has been pursued to improve peptide conformational rigidity and bioavailability. The $\chi_1$ (N-C-C-S) dihedral angle change from −66°

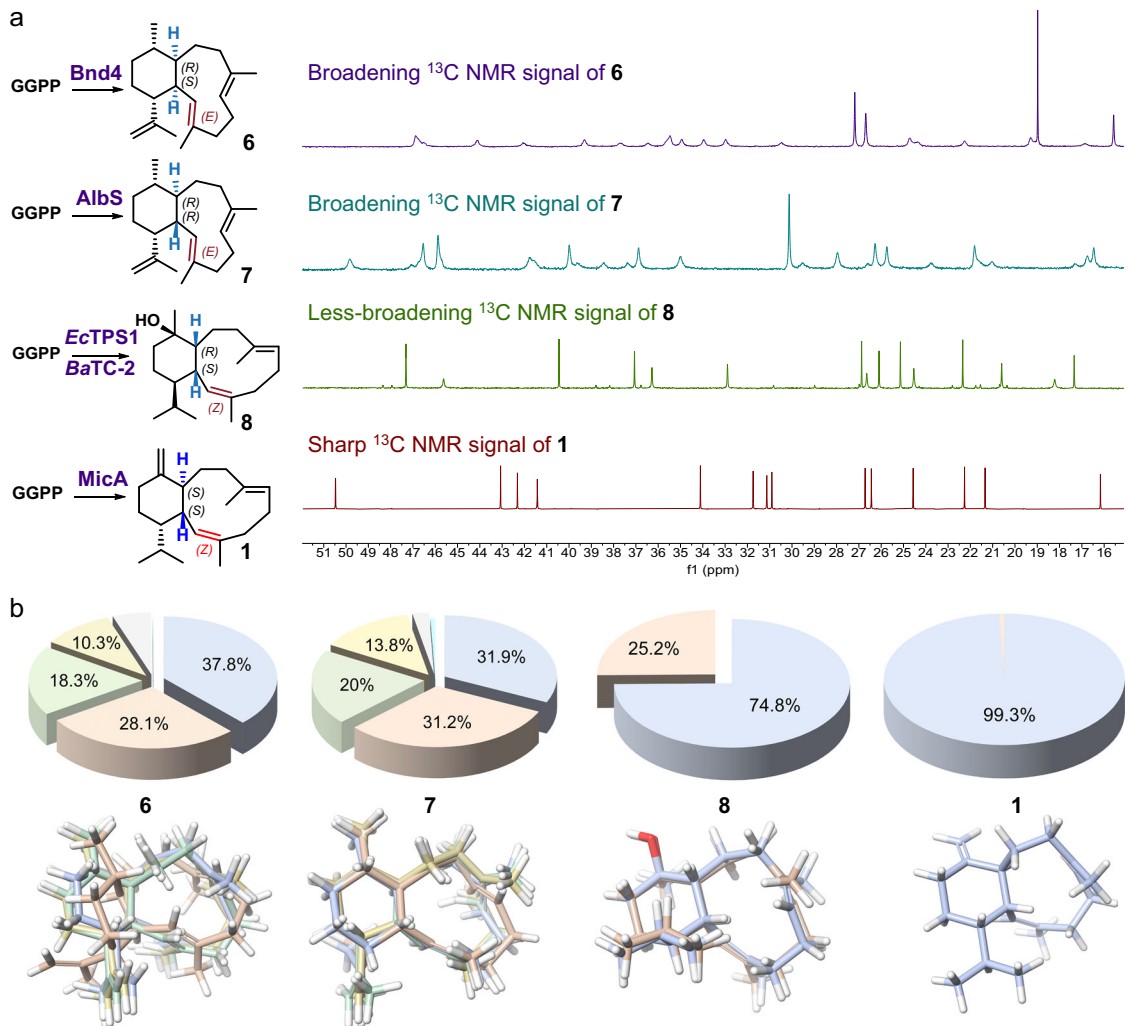

**Fig. 3 | Mechanism of flexibility and nearly non-flexibility among eunicellane scaffolds. a** [13]C NMR spectra (150 MHz) of compounds **1** and **6–8** were recorded under identical conditions using the same instrument. Each spectrum was acquired with 1024 scans and the sample concentration was standardized to 10 mg in 500 μL of deuterated benzene at 298 K. All isolated eunicellanes were confirmed to be pure through HPLC analysis (Supplementary Fig. 10). **b** The Boltzmann populations of compounds **1** and **6–8** were determined by DFT calculations at the mPW1PW91/6-31+G(d,p)//B3LYP/6-31+G(d,p) level. A pie chart representing the statistical distribution of all conformations was constructed, highlighting those with a population greater than 10%. The colors in the pie chart match the colors designated for each conformation below the charts.

(sulfoxide) to −167° (*S*-stereoisomer) renders the linker too short to maintain the α-helix.

## Catalytic route determination of MicA

We next pursued the investigations of the enzymatic mechanism of MicA. The 6,10-bicyclic eunicellane skeleton can be formed through two possible cyclization pathways. In the first pathway, a 1,10-cyclization is followed by a 1,3-hydride shift and a 1,14-ring closure. This pathway forms eunicellanes from Bnd4 and AlbS (Supplementary Fig. 20)[18,19,44] In the second pathway, a 1,14-cyclization occurs first, followed by two 1,2-hydride transfers and a 1,10-ring closure. The second pathway is followed by coral-derived eunicellane terpene synthases *Ec*TPS1 and *Ba*TC-2 (Supplementary Fig. 20)[20,21,44]. In the coral-derived enzymatic synthesis pathway, it has been hypothesized that prior to cyclization, GGPP undergoes isomerization to form geranyllinalyl diphosphate (GLPP) to explain the formation of 2*Z* double bond following consistent isomerization mechanisms with polytrichastrene synthase[45], humulene synthase mutant[46], and amorpha-4,11-diene synthase[47]. In the present study, we successfully synthesized GLPP as well as 2*Z*-GGPP (Methods and Supplementary Fig. 12) and incubated them with MicA in vitro, leading to the production of **1**

(Supplementary Fig. 21a), strongly suggesting that MicA probably follows a similar catalytic route as *Ec*TPS1. Based on these results, four possible cyclization pathways for MicA were proposed, as depicted in Fig. 4a and Supplementary Fig. 22. Pathway 1 suggests that GGPP isomerizes to GLPP or 2*Z*-GGPP, forming a carbocation which then undergoes a 1,14-cyclization, followed by two consecutive 1,2-hydride shifts, a 1,10-ring closure, and finally deprotonation. Pathway 2A and 2C initiate with a 1,10-cyclization, proceeding through two 1,2-hydride shifts, a 1,14-ring closure, and four additional 1,2-hydride shifts, culminating in deprotonation. Pathway 2B, on the other hand, initiates similarly with a 1,10-cyclization, but diverges with a 1,3-hydride shift, followed by a 1,14-ring closure, two 1,2-hydride shifts, another 1,3-hydride shift, and concludes with deprotonation. The acquisition of a 14-membered shunt product, (1*Z*, 2*Z*, 6*E*, 10*E*)-pinacene (**2**) that is probably formed by a 1,14-cyclization followed by a 1,2 or 1,3-hydride shift and subsequent deprotonation, suggested that pathway 1 is more favorable than pathway 2.

To validate this further, we conducted stable isotope labeling experiments, including experiments with deuterated[48] and fluorinated substrates[49,50]. In the deuterated experiment, 1,1-[2]H$_2$-GGPP was synthesized and incubated with MicA (Methods and Supplementary

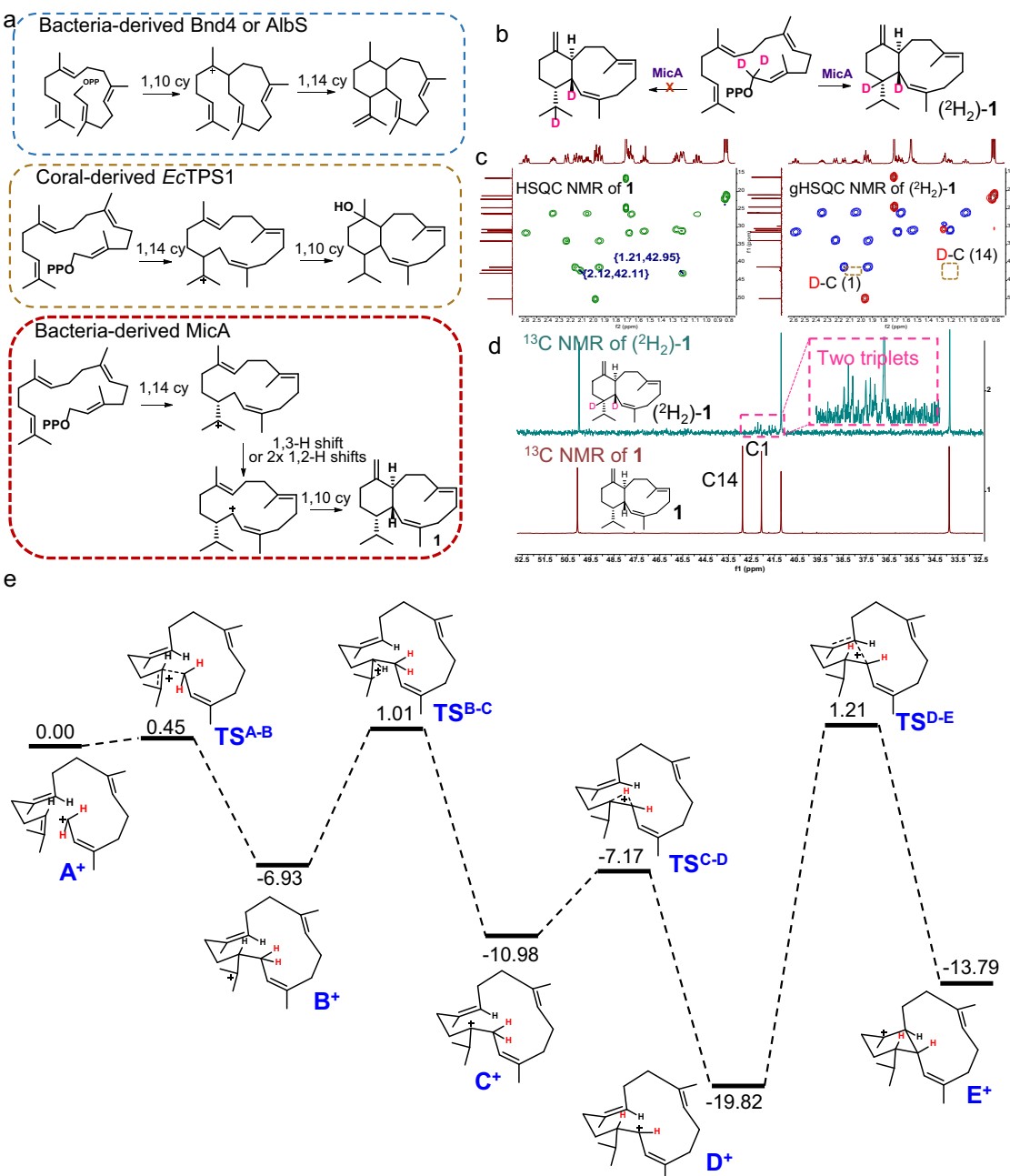

**Fig. 4 | Mechanistic investigations of MicA. a** The blue dashed box represents the cyclization route of GGPP to the 6,10-bicyclic eunicellane skeleton by bacteria-derived Bnd4 or AlbS[18,19,44]. The green dashed box represents the cyclization pathway of GGPP to the 6,10-bicyclic eunicellane skeleton by coral-derived *Ec*TPS1 (refs. [20,21,44]). The red dashed box represents the potential cyclization mechanism of GGPP to **1** by bacteria-derived MicA. Detailed hypothesis is shown in Supplementary Fig. 22. **b** Deuterium labeling experiment confirmed the occurrence of 1,2-hydride transfer during the cyclization process rather than 1,3-hydride shift. **c** HSQC spectra of **1** and (²H₂)-**1**. The green dashed box indicates the disappearance of the corresponding HSQC signals in (²H₂)-**1**. **d** ¹³C NMR spectra of (²H₂)-**1** and **1**. The pink dashed box showed a consistently decreased triplet of C1 and C14. **e** Relative free energies of intermediates and transition state structures in kcal/mol, calculated with DFT calculations utilizing the mPW1PW91/6-31+G(d,p)//B3LYP/6-31+G(d,p) method.

Fig. 12). The deuterated enzymatic product was then isolated, purified, and characterized using GC-MS and NMR. The analysis revealed that the M⁺ peak at *m/z* of 274.3 exhibited *m/z* values two Da higher than the unlabeled **1** (M⁺ peak at *m/z* of 272.2) (Supplementary Fig. 23). The ¹H NMR spectrum of (²H₂)-**1** was quite similar to that of **1**, except for the absence of signals for H-1 ($\delta_H$ 2.12) and H-14 ($\delta_H$ 1.21). Moreover, H-2 of (²H₂)-**1** displays a singlet, and H-13 exhibits a triplet of doublets splitting pattern rather than a respective doublet and a quartet of doublets pattern in **1** (Fig. 4c and Supplementary Fig. 21b–d). The corresponding carbon signals of C-1 and C-14 also show a consistently decreased triplet and slight upfield chemical shifts resulting from deuterium

isotope effects (Fig. 4d, Supplementary Figs. 40, 80 and Table 5)[51,52]. Additionally, a comparison of HSQC spectra of **1** and (²H₂)-**1** together with ²H NMR spectral analysis of (²H₂)-**1** (Supplementary Fig. 81) apparently targeted the deuterated carbons: C-1 and C-14 (Fig. 4c, Supplementary Figs. 41 and 82). Taken together, these results unanimously indicate that the deuterium atoms were retained in the product, with one deuterium atom migrating from C-1 to C-14 through two 1,2-hydride shifts instead of one 1,3-hydride shift (Fig. 4b). In addition, we chemoenzymatically synthesized 10F-GGPP (see Supplementary Information for detailed synthesis), hypothesizing that the strong electron-withdrawing effect of the fluorine atom might prevent 1,10-

cyclization, which might result in the formation of a shunt cembrane product. However, our attempt to incubate it with MicA did not yield any compounds.

To further evaluate the two hypothesized pathways, we implemented DFT calculations utilizing the mPW1PW91/6-31+G(d,p)// B3LYP/6-31+G(d,p) method[53,54]. Calculated relative free energies consistently indicated a preference for pathway 1 over pathway 2, as illustrated in Fig. 4e, Supplementary Figs. 22 and 25–30, and Supplementary Data 2. In pathway 1, a sequence unfolds that is thermodynamically favorable, with the isopropyl cation **B**$^+$ being 6.93 kcal/mol less energetic than **A**$^+$, the cembranyl cation **C**$^+$ being 4.05 kcal/mol lower than **B**$^+$, and the following cembranyl intermediate **D**$^+$ being further reduced by 8.84 kcal/mol in energy. A portion of the cembranyl cation undergoes deprotonation to form the cembrane structure (compound **2**), while the majority advances towards a further 1,10-cyclization to generate a eunicellane scaffold. However, the energy barrier for this 1,10-cyclization is 21.03 kcal/mol. It is hypothesized that the protein may enable a specific conformational change after the cembranyl cation's formation, creating a crucial intermediate conformation that aligns the C1 and C10–C11 double bond closely, thus favoring the reaction sequence towards 1,10-cyclization. Conversely, pathway 2 presents a thermodynamically less favorable process. In pathway 2 A, a 1,14-cyclization occurs following hydride shifts, with an energy barrier of 23.43 kcal/mol. Q$^+$ is predisposed to undergo deprotonation over the subsequent reaction. Pathway 2B involves a final step of 1,3-hydride shift, presenting an energy barrier of 29.94 kcal/mol. In pathway 2C, K$^+$ undergoes a 1,2-hydride shift to form L$^+$, facing a substantial energy barrier of 62.18 kcal/mol. This high energy barrier is likely due to a configurational flip that takes place during the process. Collectively, pathway 1 is identified as the energetically preferred pathway.

## Structural basis of the cyclization mechanism of MicA

Despite the unsuccessful attempts to obtain a crystal structure of MicA and the absence of structural data of homologous proteins, we managed to generate de novo predicted models of MicA using the tFold[55], RoseTTAFold[56], and AlphaFold[57] computational methods (Supplementary Fig. 31 and Supplementary data 3). Among these models, the one derived from tFold ranked first according to the "Protein Model Quality Assessment" system embedded in the tFold platform (Supplementary Fig. 32). Consequently, we used this tFold-derived model for further analysis. The overall helical structure bore a resemblance to a classical type I terpene synthase fold, and the structural alignments with Bnd4 (predicted model)[19], AlbS (predicted model)[18], and α-humulene synthase (AsR6, PDB ID: 7OC5) resulted in a root-mean-square deviation (RMSD) of 2.20 Å, 2.35 Å, and 3.88 Å, respectively (Supplementary Fig. 33). To probe the critical residues in MicA, three Mg$^{2+}$ ions were introduced to the structural model by Alphafill[58]. Subsequently, the structural model containing three Mg$^{2+}$ ions was utilized for docking with GGPP. This revealed a plausible conformation for generating **1**, with C1–C14 and C1–C10 distances of 3.87 Å and 4.62 Å, respectively (Fig. 5a and b). Similar distances correspond to those of the cyclization conformation of GGPP observed in the Bnd4 docking model[19] and CotB2 crystal structure (5GUE)[59] (Supplementary Fig. 34). We then targeted critical residues involved in eunicellane formation within 5 Å of bound GGPP in MicA. Residues of D$^{81}$D$^{82}$xxE$^{85}$ and N$^{224}$D$^{225}$xxS$^{228}$xxxE$^{232}$ motifs, which interact with the pyrophosphate group, were all confirmed to be crucial for activity through alanine mutations (Fig. 5c). Interestingly, the results showed that D$^{82}$ and E$^{85}$ residues are indispensable, which contradicts findings in Bnd4, where only the first aspartate residue is necessary. This result suggests different Mg$^{2+}$ binding forms and positions in the two eunicellane synthases. In addition, E159, a residue corresponding to E169 in Bnd4 (ref. 19), which is probably involved in assisting binding

between Mg$^{2+}$ ions and the pyrophosphate moiety, also shows a significant effect (Fig. 5c). Furthermore, residues forming the active site cavity and interacting with carbons of GGPP, such as S187, Y152, W313, W306, M74, V220, L221, G182, G183, M184, A78, F51, E53, and L54, are mainly shown in Fig. 5c. Although S187 and Y152 in MicA correspond to Y197 and F162 in Bnd4 and Y214 and Y178 in AlbS[18,19,22], alanine mutations toward these two residues did not result in the previously observed product profile change. Instead, the mutations led to primary product enhancement (S187A) or elimination (Y152A) (Fig. 5c).

It is worth noting that V220 and L221 resulted in different product profiles in alanine mutations (Fig. 5d). Purification and spectroscopic analysis allowed us to determine the new structures (**3**–**5**) as shown in Fig. 5e (see Supplementary Information for detailed elucidation process). **3** spontaneously converted into **4** during silica chromatography, a phenomenon not observed in the isolation of **1** but detected in chloroform-treated albireticulene[18]. We repurified **3** with triethylamine-treated silica resulting in no **4** being detected after purification. Accordingly, the formation of **4** is likely driven by acid-based catalysis, where the C6 double bond attacks a proton, initiating cyclization, followed by 2,7-cyclization and C20 deprotonation to generate **4** (Supplementary Fig. 35). In the new eunicellane structure of **3**, the final deprotonation occurs at C10 rather than C18. This introduces a $sp^2$ hybridized carbon into the "bridge" of the eunicellane scaffolds, aligning three carbons of the macrocycle in the same plane, which would make the macrocycle non-flexible, as justified by the sharp NMR signals of **3** (Supplementary Figs. 84 and 85). In addition, the deprotonation in **5** also follows a different direction, with a proton of C13 being eliminated. Moreover, the relative stereochemistry of C14 and C1 in **3** are different from those in **1**, indicating that a smaller residue substitution of V220 makes the bound GGPP exhibit a different pre-cyclized conformation. Significantly, all the obtained structures display *E*-configured C2 double bonds, contrasting with observations in **1** and **2**, suggesting that V220 and L221 function as *cis* and *trans* stereochemistry gatekeepers. V220 exhibits distances of 4.97 Å and 4.00 Å to C2 and C3, while V220 is in close proximity to C4 (4.65 Å) and C5 (4.24 Å) (Supplementary Fig. 36a). Considering that the requisite isomerization involved in transitioning from the 2*E* to the 2*Z* alkene via GLPP or 2*Z*-GGPP, we conjectured that they could stabilize the carbocation intermediate, there by facilitating downstream isomerization process for the generation of a 2*Z* double bond. (Supplementary Fig. 22). Carsten Schotte et al.[46] have reported similar discovery that L285 in α-humulene synthase, AsR6, can regulate the *cis* and *trans* configurations of the C2 double bond. The mutation L285M resulted in a predominant formation of 2*Z*-humulene (>85%), whereas the mutation L285A showed no significant effect on product formation. Interestingly, in our investigation, residues V220 and L221, which bear close resemblance to L285 (Supplementary Fig. 36b), did not exhibit a *cis-trans* configuration switch upon mutation to methionine residue (Supplementary Fig. 37). Conversely, when mutated to alanine, the generation of 2*E* products occurred, indicating distinctive enzymatic microenvironments and mechanisms between these two terpene synthases.

## Discussion

The current work presented the first nearly non-flexible eunicellane scaffold synthesizing enzyme, MicA, discovered by genome mining from bacteria. Through extensive mechanistic investigations, MicA was proven to synthesize an unusual eunicellane scaffold possessing a distinctive chirality combination of *trans* (C1 and C10)-*cis* ($\Delta^{2,3}$) following catalytic route as coral-derived eunicellane synthase, diverging from the bacteria-derived Bnd4 or AlbS. These discoveries have enabled us to propose a principle that a "*trans-cis*" combination gives almost inflexible eunicellane skeletons. Quantum chemical

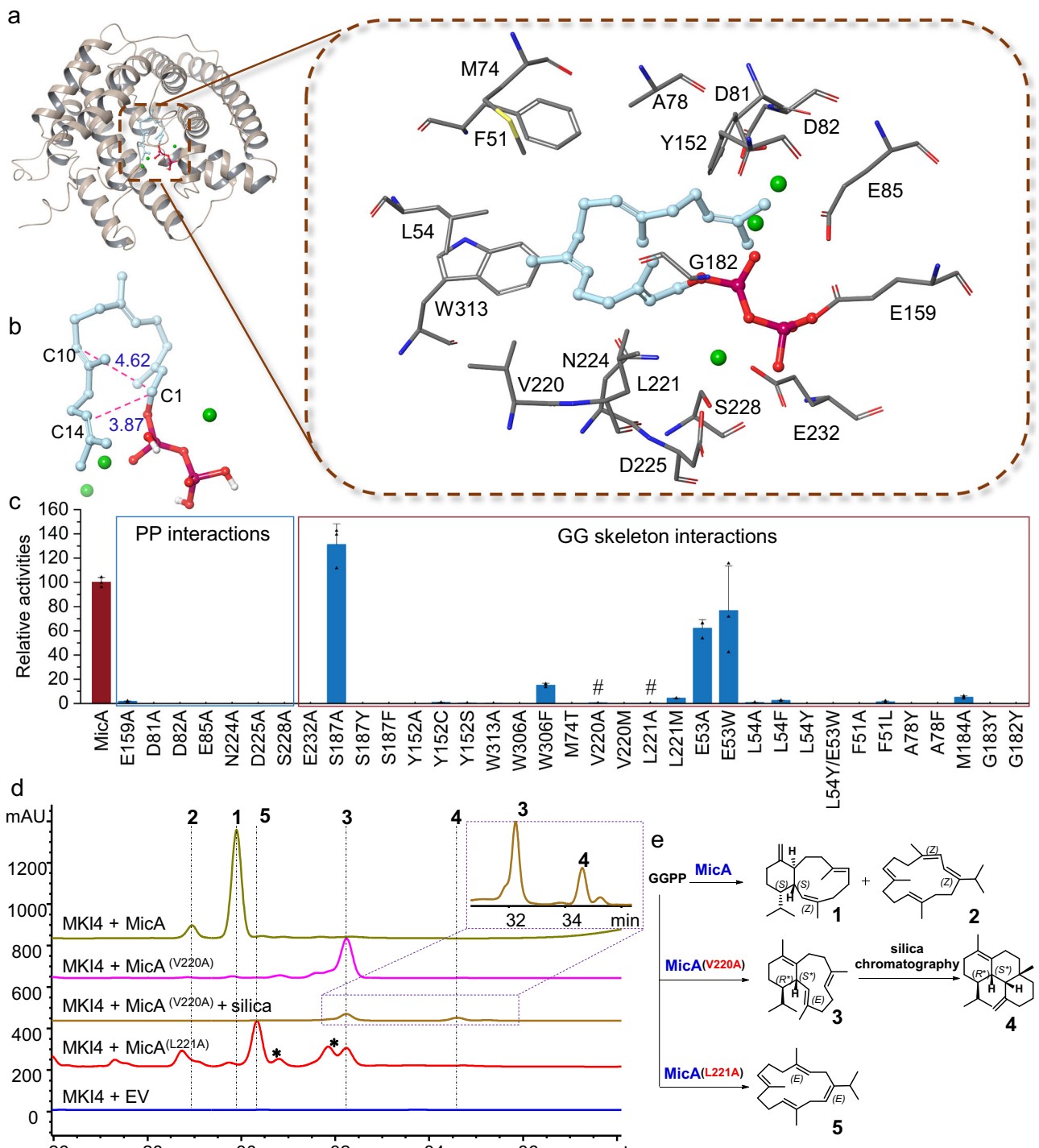

**Fig. 5 | Structural investigations of MicA. a** Investigations utilized a tFold-based model of MicA. The structural model of MicA highlights key active site residues within a 5 Å distance, and the docked GGPP, depicted in stick, exhibits a pre-cyclization conformation. **b** The docking results facilitated the measurement of distances from C1 to C10 and C1 to C14. The distances are shown in Å. **c** Comparative analysis of relative activities of wild-type MicA (red) and its mutants (blue). The yield of **1** was quantified, establishing the relative activity of wild-type MicA at 100%. Three biological parallel (*n* = 3) replicates were performed for both wild-type MicA and its mutants, with the relative activities of each replicate indicated by triangular points. The data are presented as mean values plus the standard error of the mean (SEM). Compared to MicA, all relative activities of mutants have *p* values less than 0.0001, except for MicA(E53W) with a value of 0.0025. Source data are provided as a Source Data file. The "♯"shown denotes mutants that can produce new compounds. **d** HPLC traces (210 nm) of activities of wild-type MicA, MicA(V220A), and MicA(L221A), The term "EV" refers to an "empty vector" and products marked with asterisks (*) were uncharacterized in this study. **e** The chemical structures of the diterpenes synthesized by MicA(V220A) and MicA(L221A) are presented.

calculations were conducted to investigate this proposed principle, yielding consistent Boltzmann population ratios.

Schmid et al.[28] previously observed that collinolactone manifests two distinct sets of NMR signals, occurring in a 3:1 ratio, attributed to the rotating of the methyl group attached to $\Delta^{13,14}$. Similarly, ref. 26 discovered that the configuration of $\Delta^{6,7}$ in the highly functionalized eunicellanes substantially influences conformational flipping, with (6Z) $\Delta^{6,7}$ being a less strained and more stable version. However, the (6E) $\Delta^{6,7}$

version displays broadening NMR signals and is prone to spontaneous conversion into a methoxy derivative in methanol. This observation aligns with our discovery and proposed principles, as *cis* configuration results in a less strained structure despite targeting different carbons. However, our findings contradict their discovery in a specific aspect as nearly non-flexible microeunicellene (**1**) has ($6E$) $\Delta^{6,7}$, suggesting that our "*trans-cis*" principle governs the flexibility of the carbon skeleton of eunicellane rather than their downstream functionalized counterparts.

The above information drove us to think further that besides the terpene synthases responsible for synthesizing carbon skeletons, downstream enzymes performing functionalization would also have implications for attenuating flexibility. For instance, two ether bridge bonds, a typical characteristic among coral eunicellane terpenoids, would be introduced in polyanthellin A, resulting in a dominant conformation[60,61]. However, the relevant oxidative enzymes have yet to be discovered, which presents a significant challenge for future studies.

Overall, this study reveals the discovery of an unusual eunicellane synthase, which has enabled us to identify a hidden principle that regulates the conformational changes of the macrocycle, further underscoring the significance of genome mining, biosynthesis, and enzymatic mechanistic studies.

## Methods

### Instruments and materials
NMR spectra were measured with a Bruker 600 spectrometer (Bruker Biospin AG, Fällanden, Germany) using TMS as an internal standard. Commercial silica gel (200–300 and 300–400 mesh, Qingdao Haiyang Chemical Group Co., Ltd., Qingdao, China) was used for column chromatography (CC). Reversed-phase (RP) HPLC was performed on an Agilent 1260 Infinity LC equipped with an Agilent Zorbax SB-C18 column (150 mm × 4.6 mm, 5 µm). Preparative HPLC was carried out on an Agilent 1260 Infinity LC equipped with an Agilent Eclipse XDB-C18 column (250 mm × 21.2 mm, 7 µm). All solvents used for CC and HPLC were of analytical grade (Shanghai Chemical Reagents Co., Ltd.) and chromatographic grade (Dikma Technologies Inc., CA, USA), respectively. X-ray diffraction study was carried out on a Bruker D8 Venture diffractometer. GC-MS analyses used a TRACE 1300 GC system (Thermo Scientific, Milan, Italy), equipped with a TriPlus RSH autosampler (Thermo Scientific, Switzerland), coupled to a Q Exactive GC Orbitrap mass spectrometer (Thermo Scientific, Bremen, Germany). The MS was controlled by Xcalibur® software (Version 4.1, Thermo Scientific), and the database was NIST MS Search 2.3. The enzymes needed for the cloning were sourced from TransGen Biotech. Isopropyl-*beta*-D-thiogalactopyranoside (IPTG) and isoprenol were ordered from BBI Life Sciences Corporation and Sigma-Aldrich. The information on strains, plasmids, and primer sequences has been included in Supplementary Tables 1–3, and the DNA and protein sequences of MicA are provided in Supplementary Table 4.

### Isolation of compounds 1–5
The plasmids carrying CDF-MKI4 and pET28a-MicA were transformed into *E. coli* BL21 Gold (DE3). The transformed strains were cultured in lysogeny broth (LB) containing kanamycin (50 mg/L) and streptomycin (50 mg/L). After overnight growth, the cultures were inoculated into 90 × 1 L of fresh LB medium. IPTG (0.5 mM) and isoprenol (1.0 mM) were added to the cultures when $OD_{600}$ reached 1.5 and further incubated at 28 °C and 250 rpm for an additional 18 h. After centrifuging the cultures at 3500 *g* for 15 min, the resulting pellet was extracted with methanol (MeOH). Next, the organic extract underwent partitioning with petroleum ether (PE). The resulting PE solution was concentrated under reduced pressure, resulting in a yellow oil residue weighing 1.8 g. A portion of the residue (1.0 g) was subjected to silica gel CC (300–400 mesh), yielding compound **1** (466.1 mg). Another

portion of the residue (500.0 mg) was initially fractioned by octadecylsilyl. It was further purified by RP HPLC [Acetonitrile ($CH_3CN$)/$H_2O$ (95:5)] subsequently to afford compound **2** (3.4 mg, $t_R$ = 28.9 min) and several minor peaks. At the same time, the minor peaks were directly analyzed by GC-MS. The isolation and purification for compounds **3**–**5** follow a similar procedure as the previous steps, with variations in the plasmids used. For compounds **3** and **4**, the plasmids containing CDF-MKI4 and pET28a-MicA$^{(V220A)}$ were introduced into *E. coli* BL21 Gold (DE3) for production. It is worth mentioning that compound **4** was isolated via silica gel CC from compound **3** and was not found in the PE phase during the first extraction process. The plasmids carrying CDF-MKI4 and pET28a-MicA$^{(L221A)}$ were utilized to produce compound **5**.

### Synthesis of microeunicellene derivatives (1a–1d)
Compound **1** (10.9 mg, 0.04 mmol, 1.0 eq.) was dissolved in ether (Et₂O) (1.0 mL) and treated with NaHCO₃ (6.7 mg, 0.08 mmol, 2.0 eq.) and then stirred in an ice bath. In a separate step, *m*CPBA (6.9 mg, 0.04 mmol, 1.0 eq.) was dissolved in Et₂O (1.0 mL) and added to the solution. Following 1 h of reaction, the solution was partitioned with Et₂O. The Et₂O solution was concentrated under reduced pressure and subsequently purified through silica gel CC [PE, PE/Ethylacetate (EA) (100:1 → 9:1)] to obtain compound **1a** (6.4 mg, 59% yield). Through the addition of 2 equivalents of *m*CPBA, compounds **1b** (3.2 mg, 30% yield) and **1c** (6.9 mg, 64% yield) were successfully synthesized. However, when three equivalents of *m*CPBA were used, compound **1d** (8.2 mg, 75% yield) was formed as the predominant product. Detailed structural elucidation and assignment are shown in Supplementary Information.

### GC-MS analysis
The GC-MS analyses were conducted using a TRACE 1300 GC system (Thermo Scientific, Milan, Italy), which was equipped with a TriPlus RSH autosampler (Thermo Scientific, Switzerland). This system was coupled to a Q Exactive GC Orbitrap mass spectrometer (Thermo Scientific, Bremen, Germany). The compounds and fractions prepared by reversed-phase HPLC (4 samples, Supplementary Figs. 5–9) were dissolved in acetonitrile (0.10 mg/mL) for the GC-MS analysis. These solutions were then filtered using a 0.22 µm nylon 66 syringe filter (Sigma) to obtain the test samples. The GC settings were as follows: a TraceGOLD TG-5SilMS 30 m × 0.25 mm I.D. × 0.25 µm film capillary column (Thermo Scientific, USA) was used. We injected 1 µL of sample material (with equal volumes of acetonitrile as the blank control) into the GC at a 20:1 split ratio. The inlet temperature was set at 250 °C, and a constant helium flow of 1.2 mL/min was used as the carrier gas. The gradient elution was controlled by an oven temperature program, which started at 50 °C (held for 3 min), then increased by 10 °C/min up to 280 °C (held for 5 min). This resulted in a total run-time of 31.0 min. The MS was operated in the electron ionization positive mode (70 ev) at an ion source temperature of 250 °C. The acquisition was carried out in full-scan mode within a mass range of 35–500 m/z at a resolving power of 60,000. Each sample was collected once for data acquisition. The MS data was analyzed using Xcalibur® software (Version 4.1, Thermo Scientific), and the database used was NIST MS Search 2.3.

### X-ray diffraction analysis of 1, 1a–1d
The crystallographic data were collected on a Bruker D8 Venture diffractometer equipped with Cu Kα radiation ($\lambda$ = 1.54178 Å). The crystals were kept at 171.0 K during data collection. The structures were solved with the ShelXT[62], structure solution program using Intrinsic Phasing and refined with the ShelXL[63], refinement package using Least Squares minimization.

Microeunicellene (**1**) was crystallized directly by leaving a large amount (~400 mg) in the fridge (−20 °C). Crystal data ($M$ = 272.45 g/mol): [Flack parameter: 0.60(4)], orthorhombic, space group P2₁2₁2₁ (no. 19), $a$ = 9.4505(3) Å, $b$ = 10.2972(3) Å, $c$ = 35.6882(11) Å, $V$ = 3472.95(18) Å³, $Z$ = 8, $T$ = 100 K, $\mu$ (Cu Kα) = 0.419 mm⁻¹,

$Dcalc = 1.042$ g/cm³, 115639 reflections measured ($4.952° \leq 2\theta \leq 160.484°$), 7517 unique ($R_{int} = 0.0877$, $R_{sigma} = 0.0326$) which were used in all calculations. The final $R_1$ was 0.0508 ($I > 2\sigma(I)$) and $wR_2$ was 0.1250 (all data). The crystallographic data were deposited at the Cambridge Crystallographic Data Centre with CCDC number 2310340.

(6*S*,7*S*)-Epoxy-microeunicellene (**1a**) was crystallized from MeOH at 4 °C, m.p. 44–46 °C. Crystal data ($M = 288.45$ g/mol): [Flack parameter: 0.00(7)], monoclinic, space group P2₁ (no. 4), $a = 7.9126(2)$ Å, $b = 6.8092(2)$ Å, $c = 16.4871(5)$ Å, $\beta = 93.189(2)°$, $V = 886.92(4)$ Å³, $Z = 2$, $T = 100$ K, $\mu$ (Cu K$\alpha$) = 0.479 mm⁻¹, Dcalc = 1.080 g/cm³, 1951 reflections measured ($5.368° \leq 2\theta \leq 148.862°$), 1951 unique ($R_{sigma} = 0.0538$) which were used in all calculations. The final $R_1$ was 0.0634 ($I > 2\sigma(I)$) and $wR_2$ was 0.1690 (all data). The crystallographic data were deposited at the Cambridge Crystallographic Data Centre with CCDC number 2310341.

(6*S*,7*S*,11*R*)-*Bis*-epoxy-microeunicellene (**1b**) was crystallized from PE/isopropyl alcohol (IPA) (8:2) at 4 °C, m.p. 127–129 °C. Crystal data ($M = 304.45$ g/mol): [Flack parameter: 0.09(11)], orthorhombic, space group P2₁2₁2₁ (no. 19), $a = 8.8696(4)$ Å, $b = 9.6785(4)$ Å, $c = 21.0117(9)$ Å, $V = 1803.74(13)$ Å³, $Z = 4$, $T = 170$ K, $\mu$ (Cu K$\alpha$) = 0.538 mm⁻¹, Dcalc = 1.121 g/cm³, 36301 reflections measured ($8.416° \leq 2\theta \leq 133.18°$), 3194 unique ($R_{int} = 0.0807$, $R_{sigma} = 0.0351$) which were used in all calculations. The final $R_1$ was 0.0355 ($I > 2\sigma(I)$) and $wR_2$ was 0.0909 (all data). The crystallographic data were deposited at the Cambridge Crystallographic Data Centre with CCDC number 2310342.

(6*S*,7*S*,11*S*)-*Bis*-epoxy-microeunicellene (**1c**) was crystallized from ethyl acetate (EA) at room temperature (rt), m.p. 100–102 °C. Crystal Data ($M = 304.45$ g/mol): [Flack parameter: −0.10(15)], monoclinic, space group P2₁ (no. 4), $a = 7.9978(4)$ Å, $b = 6.7005(2)$ Å, $c = 16.6441(6)$ Å, $\beta = 93.924(3)°$, $V = 889.85(6)$ Å³, $Z = 2$, $T = 100$ K, $\mu$ (Cu K$\alpha$) = 0.545 mm⁻¹, Dcalc = 1.136 g/cm³, 16279 reflections measured ($5.322° \leq 2\theta \leq 149.414°$), 3478 unique ($R_{int} = 0.0562$, $R_{sigma} = 0.0424$) which were used in all calculations. The final $R_1$ was 0.0413 ($I > 2\sigma(I)$) and $wR_2$ was 0.1079 (all data). The crystallographic data were deposited at the Cambridge Crystallographic Data Centre with CCDC number 2310347.

(2*S*,3*R*,6*S*,7*S*,11*S*)-*Tri*-epoxy-microeunicellene (**1d**) was crystallized from PE/dichloromethane (DCM) (1:1) at rt, m.p. 140–142 °C. Crystal Data for ($M = 320.45$ g/mol): [Flack parameter: −0.02(10)], orthorhombic, space group P2₁2₁2₁ (no. 19), $a = 33.6260(8)$ Å, $b = 6.7974(2)$ Å, $c = 7.9786(2)$ Å, $V = 1823.66(8)$ Å³, $Z = 4$, $T = 100$ K, $\mu$ (Cu K$\alpha$) = 0.599 mm⁻¹, Dcalc = 1.167 g/cm³, 24,595 reflections measured ($5.256° \leq 2\theta \leq 149.596°$), 3736 unique ($R_{int} = 0.0835$, $R_{sigma} = 0.0485$) which were used in all calculations. The final $R_1$ was 0.0725 ($I > 2\sigma(I)$) and $wR_2$ was 0.2221 (all data). The crystallographic data were deposited at the Cambridge Crystallographic Data Centre with CCDC number 2310348.

## Synthesis of geranyllinaloyl diphosphate (GLPP)

**1-Chloro-geranyllinalool.** Conditions for the following reaction were based on those described by Cane[64]. A solution of mixtures of (*S*)- and (*R*)-geranyllinalool (CAS No.: 1113-21-9, 435.02 mg, 1.497 mmol) and triphenylphosphine (CAS No.: 603-35-0, 785.30 mg, 2.994 mol, 2.00 eq.) in 5 mL of carbon tetrachloride (CAS No.: 56-23-5) was heated at reflux at 84 °C for 12 h. The solution was cooled and diluted with pentane (50 mL). The triphenylphosphine oxide precipitated from the solution was removed by vacuum filtration through a fritted-glass funnel. Most of the solvent was removed with a rotary evaporator at aspiratory pressure. The last traces of solvent were removed by pumping at a high vacuum for one and a half hours to yield 254.25 mg

1-chloro-geranyllinalool (55%). The resulting pale-yellow oil was used directly in the next step.

**Geranyllinaloyl diphosphate (GLPP).** Woodside's procedure was employed to prepare GLPP[65]. Tris (tetrabutylammonium) hydrogen pyrophosphate trihydrate (($Bu_4N)_3P_2O_7H$) was prepared as follows. Disodium dihydrogen pyrophosphate (CAS No.: 7758-16-9, 3.10 g, 15.349 mmol) was dissolved in 25 mL of 4% (v/v) aqueous ammonium hydroxide. The clear solution was loaded onto a 2 × 30 cm column of Dowex AG 50W-X8 cation exchange resin (100–200 mesh, H⁺ form), which had been prewashed with deionized water. The free acid was eluted with 150 mL of deionized water, and the eluant was immediately titrated to pH 7.3 with 25% (w/w) aqueous tetrabutylammonium hydroxide (CAS No.: 2052-49-5, ~15 mL). The resulting solution (~190 mL total volume) was dried by freezing the solution in a dry ice isopropanol bath and lyophilizing for ~2 days to yield a hygroscopic white solid (9.80 g, 77%), which was stored over phosphorus pentoxide until required.

The mixture of 1-chloro-geranyllinalool (0.823 mmol), ($Bu_4N)_3P_2O_7H$ (1.92 g, 2.008 mmol, 2.44 eq.) and acetonitrile (5 mL) was stirred at room temperature for 2 h. The solvent was then removed with a rotary evaporator using a 40 °C water bath to give a yellow oil, which was dissolved in ion-exchange buffer (1:49 *i*PrOH:25 mM NH₄HCO₃), and the aqueous phase was washed with Et₂O (2 × 4 mL), collected, and loaded onto a 4 × 15 cm column of Dowex AG 50W-X8 cation exchange resin (100–200 mesh, NH₄⁺ form). The column was eluted with 360 mL (two column volumes) of ion-exchange buffer, and the fractions containing GLPP (fractions that stained purple on silica TLC plates with vanillin-sulfuric acid) were lyophilized to give a white solid (to prevent moisture absorption and oxidation, store in −80 °C with nitrogen padding) containing inorganic phosphate and probably some NH₄HCO₃ (164.99 mg, ~40%). ¹H NMR (D₂O, 600 MHz) of GLPP: $\delta$ 1.55 (s, 3H, CH₃), 1.58 (s, 3H, CH₃), 1.63 (s, 3H, CH₃), 1.69 (s, 3H, CH₃), 1.90–2.14 (m, 12H, CH₂), 4.44 (m, 2H, CH₂), 5.10 (m, 3H, vinyl H), 5.43 (m, 1H).

## Synthesis of 2*Z*-geranylgeranyl diphosphate (2*Z*-GGPP) and GGPP

**Ethyl geranylgeranate.** This compound was synthesized according to a published procedure by Rabe[66]. Diisopropylamine (CAS No.: 108-18-9, 3.52 mL, 25.163 mmol, 1.05 eq.) was dissolved in dry THF (120 mL) and cooled to 0 °C. After adding *n*BuLi (CAS No.: 109-72-8, 2.7 M in hexane, 9.33 mL, 25.191 mmol, 1.05 eq.), the reaction mixture was stirred for 60 min at 0 °C. The mixture was cooled to −78 °C and triethyl phosphonoacetate (CAS No.: 867-13-0, 4.76 mL, 24.012 mmol, 1.00 eq.) was added and stirred for 2 h at −78 °C, followed by the addition of the (5*E*,9*E*)-6,10,14-trimethylpentadeca-5,9,13-trien-2-one (CAS No.: 1117-52-8, 7.14 mL, 23.982 mmol, 1.00 eq.). The reaction mixture was stirred for 2 h at −78 °C, then warmed to room temperature and stirred overnight. The reaction was quenched with 120 mL H₂O, and the mixture was extracted three times with Et₂O. The combined organic layers were dried with MgSO₄ and concentrated under reduced pressure. The residue was purified by CC on silica gel with PE/EA (200:1) to yield 7.82 g (98%) ethyl geranylgeranate as mixtures of 2*Z* and 2*E* diastereomers (1:5).

**2Z- and 2E- geranylgeraniol.** The ethyl geranylgeranate (4.60 g, 13.833 mmol, 1.00 eq.) was dissolved in dry $Et_2O$ and cooled to 0 °C. After adding DIBAlH (CAS No.: 1191-15-7, 1 M in hexane, 30.52 mL, 30.520 mmol, 2.21 eq.), the reaction mixture was stirred for 2 h. The reaction was quenched by adding 1.2 mL water dropwise, 1.2 mL 15% NaOH, and 3 mL water. The mixture was extracted three times with $Et_2O$. The combined organic layers were dried with $MgSO_4$ and concentrated under reduced pressure. The residue was purified by CC on silica gel with PE/EA (20:1) to give the 2Z- and 2E- geranylgeraniol (691.08 mg and 2.66 g, 17% and 66%). $^1$H NMR ($CDCl_3$, 600 MHz) of 2Z- geranylgeraniol: $\delta$ 1.58 (s, 3H, $CH_3$), 1.59 (s, 3H, $CH_3$), 1.59 (s, 3H, $CH_3$), 1.67 (s, 3H, $CH_3$), 1.74 (s, 3H, $CH_3$), 1.94–2.10 (m, 12H, $CH_2$), 4.08 (d, 1H, $J = 7.2$ Hz, $CH_2$), 5.09 (m, 3H, vinyl H), 5.42 (brt, 1H, $J = 7.1$ Hz, vinyl H). $^1$H NMR ($CDCl_3$, 600 MHz) of 2E-geranylgeraniol: $\delta$ 1.59 (s, 3H, $CH_3$), 1.60 (s, 3H, $CH_3$), 1.60 (s, 3H, $CH_3$), 1.68 (s, 3H, $CH_3$), 1.68 (s, 3H, $CH_3$), 1.95–2.14 (m, 12H, $CH_2$), 4.15 (d, 1H, $J = 6.9$ Hz, $CH_2$), 5.11 (m, 3H, vinyl H), 5.42 (tq, 1H, $J = 7.0$, 1.4 Hz, vinyl H).

**2Z-GGPP and GGPP.** Diphosphorylations of 2Z- and 2E- geranylgeraniol (401.12 mg and 405.25 mg) were carried out as described above for GLPP to yield 2Z-GGPP and GGPP (159.27 mg and 139.92 mg, 23% and 20%). $^1$H NMR ($D_2O$, 600 MHz) of 2Z-GGPP: $\delta$ 1.56 (s, 3H, $CH_3$), 1.56 (s, 3H, $CH_3$), 1.59 (s, 3H, $CH_3$), 1.63 (s, 3H, $CH_3$), 1.75 (s, 3H, $CH_3$), 1.91–2.15 (m, 12H, $CH_2$), 4.44 (t, 2H, $J = 7.2$ Hz, $CH_2$), 5.07 (t, 1H, $J = 8.6$ Hz, vinyl H), 5.10 (t, 1H, $J = 8.1$ Hz, vinyl H), 5.15 (t, 1H, $J = 7.0$ Hz, vinyl H), 5.46 (t, 1H, $J = 7.0$ Hz, vinyl H). $^1$H NMR ($D_2O$, 600 MHz) of GGPP: $\delta$ 1.62 (s, 3H, $CH_3$), 1.63 (s, 3H, $CH_3$), 1.64 (s, 3H, $CH_3$), 1.70 (s, 3H, $CH_3$), 1.74 (s, 3H, $CH_3$), 2.01–2.20 (m, 12H, $CH_2$), 4.48 (t, 2H, $J = 6.6$ Hz, $CH_2$), 5.20 (t, 2H, $J = 7.2$ Hz, vinyl H), 5.24 (t, 1H, $J = 7.0$ Hz, vinyl H), 5.48 (t, 1H, $J = 7.2$ Hz, vinyl H).

### Synthesis of 1,1-$^2$H$_2$-GGPP

**1,1-$^2$H$_2$-2E-geranylgeraniol.** The ethyl geranylgeranate (4.56 g, 13.713 mmol, 1.00 eq.) was dissolved in dry THF and cooled to 0 °C. After adding LiAl$^2$H$_4$ (CAS No.: 14128-54-2, 1.15 g, 27.394 mmol, 2.00 eq.), the reaction mixture was stirred for 4 h. The reaction was quenched by adding 1.0 mL water dropwise, 1.0 mL 15% NaOH, and 3.0 mL water. The mixture was extracted three times with $Et_2O$. The combined organic layers were dried with $MgSO_4$ and concentrated under reduced pressure. The residue was purified by CC on silica gel with PE/EA (20:1) to give the 1,1-$^2$H$_2$-2E-geranylgeraniol (2.76 g, 69%). $^1$H NMR ($CDCl_3$, 600 MHz) of 1,1-$^2$H$_2$-2E-geranylgeraniol: $\delta$ 1.59 (s, 3H, $CH_3$), 1.60 (s, 3H, $CH_3$), 1.60 (s, 3H, $CH_3$), 1.68 (s, 3H, $CH_3$), 1.68 (s, 3H, $CH_3$), 1.95–2.14 (m, 12H, $CH_2$), 5.10 (m, 3H, vinyl H), 5.41 (s, 1H, vinyl H).

**1,1-$^2$H$_2$-GGPP.** Diphosphorylations of 1,1-$^2$H$_2$-2E-geranylgeraniol (415.22 mg) were carried out as described above for GLPP to give 1,1-$^2$H$_2$-GGPP (135.80 mg, 19%). $^1$H NMR ($D_2O$, 600 MHz) of 1,1-$^2$H$_2$-GGPP: $\delta$ 1.56 (s, 3H, $CH_3$), 1.56 (s, 3H, $CH_3$), 1.59 (s, 3H, $CH_3$), 1.64 (s, 3H, $CH_3$), 1.70 (s, 3H, $CH_3$), 1.91–2.13 (m, 12H, $CH_2$), 5.10 (m, 3H, vinyl H), 5.41 (s, 1H, vinyl H).

### In vitro enzymatic assays of MicA

The assays were performed in 50 mM Tris-HCl, pH 8.0, containing 5 mM GGPP, 15 mM $MgCl_2$, and 20 μM TS enzyme in a total volume of 100 μL. The reactions were initiated by the addition of enzyme and incubated for 3 h at 37 °C. The reactions were quenched with 200 μL of ice-cold acetonitrile and then 50 μL of saturated NaCl solution was added to form two layers. The upper organic layer was taken for HPLC analysis. Chromatographic separation was carried out at 35 °C, with a flow rate of 1 mL/min, and an 18 min solvent gradient from 5 to 95% $CH_3CN$ in water. The linear gradient program was run as follows: 0–3 min, 5% $CH_3CN$; 3–18 min, 5–95% $CH_3CN$; 18–35 min, 95% $CH_3CN$. Enzyme products were detected by monitoring at 210 nm with a photodiode array detector. Enzyme reactions with 2Z-GGPP, GLPP or 1,1-$^2$H$_2$-GGPP were similarly carried out.

### Gene cloning and site-directed mutation

Using synthesized pET28a-MicE plasmid, we subcloned the genetic fragment of MicE into the CDF-MKI vector, which was linearized with *XhoI*, and transformed into *E. coli* Turbo (Shanghai Weidi Biotechnology) to produce plasmid CDF-MKI-MicE. The plasmid was confirmed by DNA sequencing. Overlapped PCR was used for site-directed mutagenesis of MicA, with the synthesized gene MicA as the template using TransStar FastPfu Fly DNA polymerase. After gel extraction, the PCR products were purified and inserted into a linearized pET28a (+) vector using *BamHI* and *HindIII* restriction enzymes. The resulting constructs were then transformed into *E. coli* Turbo to generate the desired plasmids and further confirmed by DNA sequencing. The plasmids with the correct sequencing were transferred into *E. coli* BL21 Gold (DE3) for expression, followed by HPLC analysis. Primer sequences used in this study were shown in Table S3.

### Protein expression and purification

Plasmids carrying MicA and its mutants were separately introduced into *E. coli* BL21 Gold (DE3). The *E. coli* strains carrying the respective plasmids were cultured in LB containing kanamycin (50 mg/mL) for antibiotic selection. Each strain was cultured in 4 × 1 L of LB at 37 °C, shaking at 200 rpm until reaching an $OD_{600}$ of 0.6. Induction of gene expression was achieved by adding IPTG (0.3 mM), followed by further cultivation of the cells at 16 °C for 16 h at 200 rpm. The cells were collected by centrifugation at 3500 g for 10 min at 4 °C. The resulting pellet was then re-suspended in cold lysis buffer (50 mM Tris-HCl, pH 8.0, containing 150 mM NaCl). Cells were disrupted using a High-Pressure Homogenizer (Scientz), followed by centrifugation at 18,000 g for 40 min at 4 °C. The desired proteins were purified through nickel affinity chromatography by applying the supernatant onto a column filled with Ni Sepharose™ 6 Fast Flow (Cytiva), washing with wash buffer (lysis buffer containing 20 mM imidazole), and eluting with elution buffer (lysis buffer containing 500 mM imidazole). The purified protein was promptly desalted using a PD-10 column (Cytiva). The protein purities were determined by analyzing them using SDS-PAGE. Each protein was divided into individual aliquots, which were then quickly frozen in liquid nitrogen and stored at −80 °C until used.

### Protein structure prediction and docking analysis

The tFold[55], RoseTTAFold[56], and AlphaFold[57] (ColabFold v1.5.3: AlphaFold2 using MMseqs2) were employed to generate de novo predicted models of MicA. To proceed, carefully follow the instructions provided on the respective individual webpage. Afterward, the generated models were uploaded onto the tFold platform to evaluate their quality. The highest-scoring model, as determined by the "Protein Model Quality Assessment" system, was then uploaded to Alphafill (https://alphafill.eu/) to obtain the MicA structure with three $Mg^{2+}$ ions, which was then docked with the ligand (Supplementary Fig. 34). The coordinates for the tFold de novo model of MicA were deposited into figshare (https://doi.org/10.6084/m9.figshare.26124145.v1). The ligand structure was drawn and subsequently converted into 3D conformation. The receptor and prepared ligand were then submitted to SwissDock and AutoDoc4 for molecular docking analysis. The default

settings were utilized for the entire procedure. The relevant websites are provided as follows:

the tFold: https://drug.ai.tencent.com/console/cn/tfold

RoseTTAFold: https://colab.research.google.com/github/sokrypton/ColabFold/blob/main/RoseTTAFold.ipynb

AlphaFold: https://colab.research.google.com/github/sokrypton/ColabFold/blob/main/AlphaFold2.ipynb

SwissDock: http://www.swissdock.ch/docking

## DFT calculations

All the DFT calculations were carried out with the Gaussian 16 package. All geometries were optimized using the B3LYP/6-31+G(d,p) method[67–70]. All stationary points were characterized by frequency calculations and reported energies include zero-point energy corrections (unscaled). IRC calculations were performed using the Local Quadratic Approximation (LQA) algorithm[71] at this same level of theory, with a maximum of 100 points (maxpoint = 100)[72,73]. The single-point energies were calculated using the mPW1PW91 method with the 6-31+G(d,p) basis set.

## Conformational search and optimization

The conformational generation and optimization of all molecules is done by Schrödinger (version 2020) and Gauss16 package. The molecules are first prepared using LigPrep with default parameter, and then sampled through Prime Macrocycle Sampling, where the sampling intensity is set to through. All generated conformations were further optimized using the B3LYP/6-31+G(d,p)[67–70] method for all atoms in vacuum. Frequency analysis was conducted at the same level of theory to verify the stationary points to be minimum or saddle points. The single-point energies were calculated using mPW1PW91/6-31+G(d,p). We calculated the free energies of different conformations of a molecule and count the Boltzmann conformational distribution of the molecule.

## Reporting summary

Further information on research design is available in the Nature Portfolio Reporting Summary linked to this article.

## Data availability

All data that support the findings of this study are present in the paper, the Supplementary Information, and figshare. Source data are provided with this paper. The protein sequences used in this study are available in the NCBI or UniProt databases under accession codes MicA (WP_205789436.1), Bnd4 (WP_239771469.1), AlbS (A0A2A2D8W5 [https://www.sciencedirect.com/science/article/pii/S2451929422006490#mmc1]), and *Ec*TPS1 (UPI41561.1). The X-ray crystallographic coordinates for structures reported in this study have been deposited at the Cambridge Crystallographic Data Centre (CCDC), under deposition numbers 2310340, 2310341, 2310342, 2310347, 2310348. These data can be obtained free of charge from The Cambridge Crystallographic Data Centre via www.ccdc.cam.ac.uk/data_request/cif. The tFold de novo model of MicA, MicA docking model and Source Data were deposited into Figshare (https://doi.org/10.6084/m9.figshare.26124145.v1, https://doi.org/10.6084/m9.figshare.26124172.v1, and https://doi.org/10.6084/m9.figshare.26117485.v1).

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

## Acknowledgements

This work was financially supported by the National Key Research and Development Program of China (No. 2022YFC2804100 to B.X. and Y.-W.G.), the Key Program of National Natural Science Foundation of China (No. 41830535 to C.-Y.W), the Taishan Scholars Program (No.tsqn202306323 to B.X.), the Shandong Laboratory Program (SYS202205 to B.X.), Shandong Provincial Natural Science Foundation (Major Basic Research Projects) (No. ZR2019ZD18 to C.-Y.W), the Shanghai Pujiang Program (23PJ1415300 to B.X.), and the start-up funding provided by the Shanghai Institute of Materia Medica, Chinese Academy of Sciences (to B.X.). We thank Congcong Guo and the Pharmaceutical Analysis and Quality Research Platform of Bohai Rim Advanced Research Institute for Drug Discovery for all the technical support in NMR data collection. We thank Zaiyong Zhang and Pharmaceutical Quality Control and Solid-state Chemistry Research Center for all the technical support in X-ray diffraction. We thank Wenzhen Mi, Pengjun Cai, the Pharmaceutical Analysis and Quality Research Platform of Bohai Rim Advanced Research Institute for Drug Discovery, and the Institutional Technology Service Center of Shanghai Institute of Materia Medica for all the technical support in MS experiments. We appreciate *Prof*. Jeffrey D. Rudolf at the University of Florida for sharing the MKI and MKI4 plasmids, for his valuable suggestions on the project, and for his assistance with the English language improvements. We thank the Open Studio for Druggability Research of Marine Natural Products, National Laboratory for Marine Science and Technology (Qingdao, China) Directed by Kai-Xian Chen and Yue-Wei Guo. B.X. would also like to express a debt of gratitude to *Prof*. Youli Xiao at the CAS Center for Excellence in Molecular Plant Sciences, who passed away on January 30, 2020, for his previous invaluable, hard-working efforts in mentoring and training B.X.

## Author contributions

The conceptualization was done by B.X. The experimental investigation was carried out by J.L., B.C., Z.F., J.M., L.L., and X.C. The data analysis was done by J.L., B.C., Z.F., J.M., M.Z., C.-Y.W., C.W., Y.-W.G., and B.X. The manuscript was written by J.L., B.C., Z.F., J.M., C.-Y.W., C.W., Y.-W.G., and B.X. The fundings were acquired by Y.-W.G., C.-Y.W. and B.X. The project administration and supervision were carried out by C.-Y.W., C.W., Y.-W.G., and B.X.

## Competing interests

The authors declare no competing interests.
