## [Peer Review File · Nature Communications]

Discovery of a terpene synthase synthesizing a nearly non-flexible eunicellane reveals the basis of flexibilityREVIEWER COMMENTS

Reviewer #1 (Remarks to the Author):

Li et al. have reported the biosynthesis of eunicellane diterpenoids, a class of bioactive compounds with a distinct 6,10-bicycle structure, found in marine corals and some bacteria and plants. The research explores investigates the mechanisms underlying the structural flexibility of eunicellane's intrinsic macrocycle. This paper presents the discovery of a terpene synthase, MicA, which synthesizes a rigid eunicellane skeleton, challenging the current understanding of flexibility in eunicellane structures. The study suggests that the configuration of bridging carbons and adjacent double bonds determines this flexibility. Isotopic labeling and density functional theory calculations demonstrate that bacterial MicA and coral-derived synthase share similar catalytic pathways. AI-aided structural analysis and mutagenesis experiments highlight a novel enzymatic mechanism producing a rigid eunicellane scaffold. Comprehensive studies, including DFT calculations, further show that a trans-fused bicycle structure with a 2Z-configured alkene yields a non-flexible eunicellane skeleton. This research unveils a new eunicellane synthase and advances the theoretical understanding of structural rigidity in eunicellane skeletons. However, substantial revisions are required in the computational chemistry section.

Comment

1.

The computational level of B3LYP/6-31G* is inappropriate for the calculations of cationic species such as carbocations, proton etc., and thus a recalculation is required. Generally, it is necessary to include polarization functions in the calculations for cationic intermediates/transition states, so structural optimization should be performed with a basis set of 6-31G** or higher. This issue has been pointed out in the 2006 review by Matsuda et al. (Matsuda, S. P. T.; Wilson, W. K.; Xiong, Q. *Org. Biomol. Chem.* 2006, 4, 530–543.) Moreover, the optimal computational level for structural optimization differs from that for energy calculation. B3LYP is not suitable for calculating the energy of terpene cyclization reactions, which involve significant enthalpy changes. Although B3LYP/def2-TZVP is used for single-point calculations, simply changing a higher basis set is insufficient. In DFT calculations, there is a compensation between the errors arising from the functional and the

basis set, so it is necessary to use the optimal combination of functional and basis set. Upon revision, please reconsider the computational level with reference to benchmark studies and if you choose to use a different computational level than what is established in the literature, provide a benchmark to justify it.

2.

Terpene cyclization reactions often involve concerted reactions that include multiple chemical events. Therefore, for a journal such as Nature Communications, it is necessary to show all IRC plots in the Supporting Information (SI).

Reviewer #2 (Remarks to the Author):

The study is about the investigation on the rigidity of the eunicellane scaffold. This manuscript includes many experiments and theoretical studies, such as the genome mining study for the identification of a terpene synthase MicA producing a non-flexible eunicellane diterpenoid, isotopic labeling experiments, site-directed mutagenesis, and DFT calculation. The authors seem to have put in considerable effort and deserve respect for the enormous amount of work.

However, this work does not contain sufficient insight with broad appeal unfortunately. The authors emphasize the non-flexible feature of MicA product, and found that trans-configured C1 and C10 with a 2Z double bond makes the eunicellane scaffold rigid. However, they did not explain why the trans-configured C1 and C10 with a 2Z double bond can reduce the flexibility. Thus, no lessons can be learned that can be broadly applied to chemistry or other fields. Actually, it is not surprising that each stereo and cis-trans isomer of the eunicellane scaffold exhibits a different degree of flexibility or rigidity. Additionally, the flexibility of eunicellane scaffold is a quite specific topic and would attract the interest of only limited people.

In summary, this manuscript should be suitable for other journals for the specialists of natural product chemistry. In addition, there are some flaws especially in DFT calculation as shown below:

Major

(DFT calculation)

- Fig 3e, path ii and Extended Data Fig 1: The authors calculated the pathway in which one 1,3-H shift occurs when the intermediate H⁺ is converted to the intermediate I⁺. However, it is inconsistent with the author's isotopic labeling experiment using (1,1-2H₂)-GGPP as a substrate (Fig 3b). Not one 1,3-H shift but two 1,2-H shifts should be calculated (The hydrogen atom on C14 move to C15. Then, the hydrogen atom on C1 move to C14).

- The structure of the transition state between the intermediates I⁺ and E⁺ seems to be inappropriate. (I checked the structure of this transition state based on the cartesian coordinates in Table S21.) From this transition state, not E⁺ but an epimer of E⁺ in which the stereochemistry at the C1 position is inverted would be formed.

(NMR analysis to evaluate the flexibility of the eunicellane scaffold)

- Fig5a: The authors seem to obtain NMR spectra at only one temperature, and it is not shown at what temperature the authors carry out the NMR measurement. It is recommended to carry out VT (variable temperature) NMR measurement to confirm the rigidity of the MicA product.

Minor

- Figure S10: The stereochemistry at C11 position is not shown in the intermediates G⁺, H⁺, and I⁺, and the transition state between the intermediates H⁺ and I⁺. The stereochemistry at C14 position is not shown in the transition state between the intermediates I⁺ and E⁺. The stereochemistry at C10 position is not shown in the intermediate G⁺. It is recommended to show them. In addition, the double bond between C10 and C11 should be fixed to the single bond, in the structure of the transition state between the intermediates G⁺ and H⁺.

- line93: The details of "the obtained bacterium" should be provided. Which strain did the authors analyze? How did the authors get "the obtained bacterium"?

- Fig 2b: The authors should indicate what the "MKI" is, even though they show what the

“MK14” is.

- line 194: The authors just say that the anticipated compounds could not be obtained, when 10F-GGPP was used as a substrate. It should be clarified whether the authors obtained no product or obtained unanticipated products.

- line 239 “...3.82 Å distance to C17...”: C17 should be fixed to C18.

- Fig 4c: V220A mutant and L221A mutant exhibit the enzyme activity as shown in Fig 4d. However, the relative activities of these mutants are almost zero in Fig 4c. Why?

- Figure S5: The authors suggest that there are several minor products (lines 109–111, and 119–121), but only one mass spectrum is shown in Figure S5. It is recommended to add the mass spectra of the other minor products.

Reviewer #3 (Remarks to the Author):

Review on the manuscript titled “Discovery of a terpene synthase synthesizing a non-flexible eunicellane reveals the basis of flexibility”

Li et al identified a novel terpene synthase, MicA, catalyzing the formation of a less flexible new eunicellane. The new eunicellane structure has been elucidated by isotope labelling and NMR analysis. The cyclization mechanism of MicA has been proposed and confirmed by both deuterium labelling experiment and DFT calculation. The key issue lies in the structural analysis. First, the docking simulations appear to have omitted a crucial element: the metal ion, Mg²⁺. The metal ion contributes greatly to the leaving of diphosphate group, interaction with the carbocation intermediates and coordinating with several key active-site residues including aspartate (D81, D82), glutamate (E85, E232), and asparagine (N224) etc. The absence of Mg²⁺ in the docking analysis could lead to an inaccurate placement of the substrate within the binding pocket. Furthermore, including such analysis may provide valuable insights into the roles of V200 and L221. Current explanation in lines 256-270 is rather cursory. To improve the robustness of the structural analysis, a more comprehensive

approach incorporating Mg²⁺ is necessary.

Second, the V220A and L221A mutation led to the formation of two structurally different diterpene scaffolds. This observation is very interesting, however, the current explanation is not deep enough. A more detailed analysis and discussion of the implications of these mutations on the enzyme's catalytic mechanism and product formation are needed.

Major comments

1. Docking and structural analysis should be improved by incorporating the metal ion Mg²⁺ into the system.
2. More in depth analysis of the impact of V220A and L221A on the product formation.
3. In Figure 1, and Line 68, the background information about flexibility of eunicellane scaffolds should be briefly explained. It was confusing at beginning to understand the flexibility till the last section of the manuscript. As such, for better clarity, the paper structure might be re-arranged by shifting the section "Investigations of the effects of stereochemistry on flexibility among eunicellanes" in front of the section "Catalytic route determination...".

Minor comments

1. In Figure 3e and extended Data Figure 1, the calculation is based on 1,3H transfer mechanisms, will be possible to have two 1,2H transfers instead of one 1,3H transfer? In that case, the energy might be lower.
2. The structures of MicA, Bnd4, AlbS have been analysed in extended Data Figure 4. With these structures, some possible mechanisms may be proposed on the 1,10 or 1,14 cyclization of GGPP as proposed in Figure 3a.
3. Although compound 1 is definitely much more rigid than others, based on the DFT calculation, compound 1 still has small degree of flexibility. As such, is it accurate to define compound 1 as non-flexible? If not, the title may also need to be revised.

Point-by-point Response

We express our gratitude to the reviewers for their thorough review and evaluation of our manuscript. We greatly value the constructive criticism and feedback provided. In the following sections, we address each comment in detail, point by point.

Reviewer #1

1. The computational level of B3LYP/6-31G* is inappropriate for the calculations of cationic species such as carbocations, proton etc., and thus a recalculation is required. Generally, it is necessary to include polarization functions in the calculations for cationic intermediates/transition states, so structural optimization should be performed with a basis set of 6-31G** or higher. This issue has been pointed out in the 2006 review by Matsuda et al. (Matsuda, S. P. T.; Wilson, W. K.; Xiong, Q. *Org. Biomol. Chem.* 2006, 4, 530–543.)

Moreover, the optimal computational level for structural optimization differs from that for energy calculation. B3LYP is not suitable for calculating the energy of terpene cyclization reactions, which involve significant enthalpy changes. Although B3LYP/def2-TZVP is used for single-point calculations, simply changing a higher basis set is insufficient. In DFT calculations, there is a compensation between the errors arising from the functional and the basis set, so it is necessary to use the optimal combination of functional and basis set. Upon revision, please reconsider the computational level with reference to benchmark studies and if you choose to use a different computational level than what is established in the literature, provide a benchmark to justify it.

Response: Thanks for your professional suggestion. According to your suggestion, we meticulously examined the relevant literatures, including a notable review by Prof. Matsuda and colleagues (*Org. Biomol. Chem.*, 2006, 4, 530–543) and another review by Prof. Dean J. Tantillo (*Nat. Prod. Rep.*, 2011, 28, 1035–1053). For these DFT calculations, we have performed recalculation for all the carbocation-related pathways at the mPW1PW91/6-31+G(d,p)//B3LYP/6-31+G(d,p) level. In addition, pathway ii has been revised based on the reviewers' suggestions. Relevant results can be found in Figures 4e, S19–S21, and Tables S19–S20.

2. Terpene cyclization reactions often involve concerted reactions that include multiple chemical events. Therefore, for a journal such as Nature Communications, it is necessary to show all IRC plots in the Supporting Information (SI).

Response: We are grateful of your kind suggestion and all IRC plots have been shown in Figures S20 and S21.

Reviewer #2:

1. The study is about the investigation on the rigidity of the eunicellane scaffold. This manuscript includes many experiments and theoretical studies, such as the genome mining study for the identification of a terpene synthase MicA producing a non-flexible eunicellane diterpenoid, isotopic labeling experiments, site-directed mutagenesis, and DFT calculation. The authors seem to have put in considerable effort and deserve respect for the enormous amount of work.

Response: We deeply appreciate the recognition of the considerable effort we have invested in our research. It is gratifying to see our dedication acknowledged. Inspired by your valuable feedback, we are dedicated to further refining the clarity and significance of our findings. Thank you for your careful and considerate evaluation of our efforts.

2. However, this work does not contain sufficient insight with broad appeal unfortunately. The authors emphasize the non-flexible feature of MicA product, and found that *trans*-configured C1 and C10 with a 2Z double bond makes the eunicellane scaffold rigid. However, they did not explain why the *trans*-configured C1 and C10 with a 2Z double bond can reduce the flexibility. Thus, no lessons can be learned that can be broadly applied to chemistry or other fields. Actually, it is not surprising that each stereo and *cis-trans* isomer of the eunicellane scaffold exhibits a different degree of flexibility or rigidity. Additionally, the flexibility of eunicellane scaffold is a quite specific topic and would attract the interest of only limited people.

Response: We are thankful for your insightful feedback on our work. Your suggestions have significantly improved our current revision. We have addressed the two concerns you raised as follows:

(1) Clarifications on how the *trans*-configuration of C1 and C10 with a 2Z double bond contributes to the reduced flexibility of the 6,10-bicyclic eunicellane:

In this study, the Boltzmann populations of compounds **6–8**, **1**, and the hypothesized 1*S*,10*R*-microeunicellene (1*S*,10*R*-**1**), 1*R*,10*S*-microeunicellene (1*R*,10*S*-**1**), 2*E*-microeunicellene (2*E*-**1**), and 1*R*,10*R*-microeunicellene (1*R*,10*R*-**1**) (Figure 3b, Extended Data Figs. 2, 3) were computed using DFT calculation. The results showed that the conformational space of **6** is primarily occupied by four conformations, accounting for 93.8% of the total space. Similarly, for **7**, four conformations dominate the conformational space, representing 96.9% of the total space. Moreover, **8** exhibits less flexibility, with its conformational space primarily populated by two conformations. In the case of **1**, a single conformation overwhelmingly dominates the conformational space at 99.3%, exhibiting almost inflexibility. (Figure 3b, Extended Data Fig. 2). However, 1*S*,10*R*-**1** and 1*R*,10*S*-**1** exhibit flexibility when C-1 and C-10 are in a *cis* configuration, presenting two conformations (76.1% and 23.9%) for the former and three conformations (75.4%, 10.4% and 9.7%) for the latter. Similarly, 2*E*-**1** also displays flexibility when (2*Z*) $\Delta^{2,3}$ becomes (2*E*) $\Delta^{2,3}$, with two predominant conformations (80.3% and 11.2%). These findings emphasize the importance of the configurations of $\Delta^{2,3}$, C1, and C10 in determining the rigidity of the eunicellane scaffold. Specifically, the combination of a *trans*-configured C1 and C10 with a 2Z double bond leads to a dominant eunicellane conformation. Additionally, the calculated results of the hypothesized 1*R*,10*R*-**1** align with those for **1**, suggesting that the *trans* configuration of C1 and C10, regardless of the absolute orientations, has the same effect on flexibility (Extended Data Fig. 3). The distinction between conformation 1 (99.3%) and conformation 2 (0.7%) of **1** resides within the hexatomic ring. Metadynamics simulations indicated that ΔG between them is approximately 3 kcal/mol, with a corresponding energy barrier of about 7.5 kcal/mol (Supplementary Figure S13).

Conformation-1 is capable of converting into conformation-2, yet it predominantly retains conformation-1 as its primary state. This observation is in accordance with our findings from variable temperature NMR spectroscopy. With the temperature rising ($\geq 328\text{K}$), the spectrum broadens, indicating a shift in conformation and thereby revealing the presence of multiple conformations coexisting or undergoing rapid transformation. (Figures S14 and S15). Upon detailed examination of the conformations of the remaining compounds (excluding **1**), it has been observed that their primary difference stems from the rotatable bond at C7–C19, which is attached to (6*E*) $\Delta^{6,7}$ (Extended Data Figs. 2 and 3). Therefore, we hypothesized that the combination of a *trans*-configured C1 and C10 with a 2*Z* double bond restricts the rotation at C7–C19, leading to a dominant eunicellane conformation, which aligned with previous studies (*Angew. Chem. Int. Ed.* 2021, 60, 14163–14170; *Helv. Chim. Acta.*, 2000, 83, 1561–1575; *Angew. Chem. Int. Ed.*, 2021, 60, 23212–23216). We further hypothesized that the *Z* configuration of $\Delta^{2,3}$ limits the rotations of C1–C2 and C3–C4 bonds. The configurations of C1 and C10 might impact the rotation of the C1–C10 bond, which in turn influences the rotation of adjacent bonds. This cascade of effects leads to conformational changes, notably affecting the rotation of the C7–C19 bond. Moreover, the *trans* configuration of C1 and C10 could potentially offer enhanced thermodynamic stability.

(2) Broad impact of this study:

Previously, many researchers have mentioned the flexible characteristic of macrocycle-containing compounds. For instance, Ines Mancini et al. (*Helv. Chim. Acta.*, 2000, 83, 1561–1575) previously discovered that the configuration of C6 in the highly functionalized eunicellanes substantially influences conformational flipping, with (6*Z*) $\Delta^{6,7}$ being a less strained and more stable version. Similarly, Julian C. Schmid et al (*Angew. Chem. Int. Ed.*, 2021, 60, 23212–23216) reported that collinolactone shows two conformers in NMR spectra due to the rotating of the methyl group attached to (13*E*) $\Delta^{13,14}$. However, there has been a lack of in-depth research into why and how the structural geometries influence the conformations of these flexible compounds.

The study of MicA not only elucidates its enzymatic mechanism and contributes to the

enrichment of eunicellane biosynthesis, thereby offering insights for future eunicellane biosynthetic endeavors, but also aids in the structural analysis of natural products. Under the condition of assured compound purity, the nuclear magnetic resonance (NMR) spectrum reveals disproportionate integration broadening, a finding that can be referenced in the relevant research discussed in this paper.

To date, only four eunicellane terpene synthases have been documented including two reports we have participated (*Angew. Chem. Int. Ed.*, 2021, 60, 23159–23163; *Nat. Chem. Biol.*, 2022, 18, 664–669; *Chem*, 2023, 9, 698–708). The current study significantly broadens the scope of research into the biosynthesis of eunicellanes, providing a comprehensive expansion of the study's findings on MicA and further enhancing our understanding of this fascinating catalytic mechanisms.

2. Fig 3e, path ii and Extended Data Fig. 1: The authors calculated the pathway in which one 1,3-H shift occurs when the intermediate H^+ is converted to the intermediate I^+ . However, it is inconsistent with the author's isotopic labeling experiment using (1,1- $2H_2$)-GGPP as a substrate (Fig 3b). Not one 1,3-H shift but two 1,2-H shifts should be calculated (The hydrogen atom on C14 move to C15. Then, the hydrogen atom on C1 move to C14).

Response: We are grateful for your professional suggestion. We corrected our pathway ii and performed recalculations. The results have been updated in Figure 4e, Extended Data Fig. 5, and Figure S19.

3. The structure of the transition state between the intermediates I^+ and E^+ seems to be inappropriate. (I checked the structure of this transition state based on the cartesian coordinates in Table S21.) From this transition state, not E^+ but an epimer of E^+ in which the stereochemistry at the C1 position is inverted would be formed.

Response: Thank you for thorough examining the structural details of the transition state between intermediates I^+ and E^+ . The transition state (TS^{I-E}) appears to be exceptionally inappropriate, potentially owing to the less-than-ideal use of

computational parameters employed in our initial analysis. We conducted a comprehensive review of the literature, specifically the work by Prof. Dean J. Tantillo (*Nat. Prod. Rep.*, 2011, 28, 1035-1053) and recalculated all pathways at the mPW1PW91/6-31+G(d,p)//B3LYP/6-31+G(d,p) level. In addition, pathway ii has been revised in light of the reviewers' suggestions and the insights gained from our deuterium labeling experiments. The recalculated results have successfully resolved the issues.

4. (NMR analysis to evaluate the flexibility of the eunicellane scaffold)

- Fig5a: The authors seem to obtain NMR spectra at only one temperature, and it is not shown at what temperature the authors carry out the NMR measurement. It is recommended to carry out VT (variable temperature) NMR measurement to confirm the rigidity of the MicA product.

Response: We appreciate your kind suggestions. Our NMR measurements were conducted at 298K for data in Figure 3e and Figure S11. Variable temperature (VT) NMR measurements for eunicellane products **1**, and **6–8** were performed in the range of 298K–348K. With the temperature increase ($\geq 328\text{K}$), the spectra of **1**, **7**, and **8** exhibited broadening, while **6** displayed no noticeable changes (Figures S14 and 15). The slightly broaden spectra observed in **1** when the temperature increased up to 328K is in accordance with the computational results of our metadynamics simulations that ΔG between two main conformers is approximately 3 kcal/mol, with a corresponding energy barrier of about 7.5 kcal/mol (Figure S13). In addition, reviewer-3 also gave similar suggestion in the question-6. Accordingly, we have revised the “non-flexible” to “almost inflexible” or “nearly non-flexible” throughout our manuscript for accuracy.

5. Figure S10: The stereochemistry at C11 position is not shown in the intermediates G^+ , H^+ , and I^+ , and the transition state between the intermediates H^+ and I^+ . The stereochemistry at C14 position is not shown in the transition state between the intermediates I^+ and E^+ . The stereochemistry at C10 position is not shown in the intermediate G^+ . It is recommended to show them. In addition, the double bond between C10 and C11 should be fixed to the single bond, in the structure of the transition state

between the intermediates G⁺ and H⁺.

Response: We appreciate your comment and regret the oversight. Following the recommendations of the first reviewer, we have rerun the calculations for this section at the mPW1PW91/6-31+G(d,p)//B3LYP-D3BJ/6-31+G(d,p) level and have carefully analyzed the updated results. All the stereochemistry has now been included, and the double bond between C10 and C11 in TS^{G-H} (now renamed TS^{H-I}) has been corrected to a single bond, as shown in Figure S19.

6. Line93: The details of “the obtained bacterium” should be provided. Which strain did the authors analyze? How did the authors get “the obtained bacterium”?

Response: Thank you for your suggestion. The “the obtained bacterium” mentioned in Line93 was *Streptomyces albogriseolus* SY67903 (*J. Nat. Prod.*, 2020, 83, 1641–1645). We acquired the strain from one of the authors of the paper, then proceeded to genome sequencing analysis. Unfortunately, we were unable to identify the relevant biosynthetic gene cluster within this strain. The biosynthetic gene cluster in this manuscript is derived from *Micromonospora* sp. HM134 and the genome of this strain is publicly available in the NCBI database. Relevant information has been added to the Figure S3.

7. Fig 2b: The authors should indicate what the “MKI” is, even though they show what the “MKI4” is.

Response: Thank you for your professional suggestions. More information about this system has been included in the revision. For your convenience, we have added the information below:

The "MKI" expresses three genes that convert isoprenol into DMAPP, where "MKI" is an abbreviation for these three genes. “M” refers to a kinase, hydroxyethylthiazole kinase (ThiM) from *E. coli*; “K” refers to a kinase, isopentenyl phosphate kinase (ipk) from *Arabidopsis thaliana*; “I” refers to isopentenyl diphosphate isomerase (idi) from *E. coli*. The "MKI4" encodes for four genes responsible for converting isoprenol into GGPP, with "MKI" denoting an abbreviation for the three genes previously mentioned.

The "4" signifies the capability of the recombinant strain to produce four times the isoprene units. In another word, a GGPP synthase (bnd3) has been introduced in the “MKI” system for generating C20 polyprenyl pyrophosphate substrate. For your convenience, we have included a schematic overview of our terpene biosynthesis system as below:

8. Line 194: The authors just say that the anticipated compounds could not be obtained, when 10F-GGPP was used as a substrate. It should be clarified whether the authors obtained no product or obtained unanticipated products.

Response: Thank you for your comment regarding the lack of clarity on line 194 (line 266 in the revision). No product was observed when 10F-GGPP was used as a substrate. We have corrected the description in the manuscript.

9. Line239 “...3.82 Å distance to C17...”: C17 should be fixed to C18.

Response: Thank you for your thorough review and for pointing out our oversight. We agree with your observation that the reference to “C17” on line 239 should indeed be corrected to “C18” to accurately reflect the distance measurement. Further, following the third reviewer's suggestion, three Mg²⁺ ions were introduced to the structural model by Alphafill (*Nat. Methods*, 2023, 20, 205–213). Subsequently, the structural model containing three Mg²⁺ ion was utilized for docking with GGPP. The result showed that W77 was distant from C18. Consequently, we have omitted the description of the respective section from our manuscript.

10. Fig4c: V220A mutant and L221A mutant exhibit the enzyme activity as shown in Fig 4d. However, the relative activities of these mutants are almost zero in Fig 4c. Why?

Response: We sincerely thank you for careful reading. The main product of MicA is compound **1**. Thus, we evaluated the activities of MicA mutants by quantifying the yields of compound **1**. The relative activities of these mutants are nearly zero, indicating that they did not produce compound **1**. The V220A mutant and L221A mutant exhibit enzyme activity to produce compounds **3–5**, as depicted in Fig 5d, rather than compound **1**. To avoid confusion, we have added a pound sign (#) for mutants that can produce new products. A brief description of the relative activity in the legend of Figure 5c has also been included for better understanding.

11. Figure S5: The authors suggest that there are several minor products (lines 109–111, and 119–121), but only one mass spectrum is shown in Figure S5. It is recommended to add the mass spectra of the other minor products.

Response: Thanks for your careful reading. The mass spectra of the minor products have been added in Figures S5–S10.

Reviewer #3:

1. Docking and structural analysis should be improved by incorporating the metal ion Mg^{2+} into the system.

Response: Thanks for your professional suggestion. Based on your professional suggestion, three Mg^{2+} ions were introduced to the structural model by Alphafill (*Nat. Methods*, 2023, 20, 205–213). Subsequently, the structural model containing three Mg^{2+} ions was utilized for docking with GGPP. The updated docking results have been included in Figure 5a and 5b.

2. More in depth analysis of the impact of V220A and L221A on the product formation.

Response: We are grateful for your kind suggestions. V220 exhibits distances of 4.97

Å and 4.00 Å to C2 and C3, while V220 is in close proximity to C4 (4.65 Å) and C5 (4.24 Å) (Extended Data Fig. 9a). Considering that the requisite isomerization involved in transitioning from the *2E* to the *2Z* alkene via GLPP or *2Z*-GGPP, we conjectured that they could stabilize the carbocation intermediate, there by facilitating downstream isomerization process for the generation of a *2Z* double bond. (Extended Data Fig. 5). Carsten Schotte et al. have reported similar discovery that L285 in α -humulene synthase, AsR6, can regulate the *cis* and *trans* configurations of the C2 double bond (*Angew. Chem. Int. Ed.*, 2021, 60, 20308–20312). The mutation L285M resulted in a predominant formation of *2Z*-humulene (>85%), whereas the mutation L285A showed no significant effect on product formation. Interestingly, in our investigation, residues V220 and L221, which bear close resemblance to L285 (Extended Data Fig. 9b), did not exhibit a *cis-trans* configuration switch upon mutation to methionine residue (Figure S24). Conversely, when these two residues were mutated to alanine, the generation of *2E* products occurred, indicating distinctive enzymatic microenvironments and mechanisms between these two terpene synthases.

3. In Figure 1, and Line 68, the background information about flexibility of eunicellane scaffolds should be briefly explained. It was confusing at beginning to understand the flexibility till the last section of the manuscript. As such, for better clarity, the paper structure might be re-arranged by shifting the section “Investigations of the effects of stereochemistry on flexibility among eunicellanes” in front of the section “Catalytic route determination...”.

Response: Thank you for your valuable feedback on the organization of our manuscript. We revised the manuscript to provide a brief explanation of the flexibility of eunicellane scaffolds (lines 68-80 in the revision), ensuring that readers have a foundational understanding from the outset. Furthermore, we reorganized the manuscript structure as your suggestion. We moved the section “Investigations of the effects of stereochemistry on flexibility among eunicellanes” to an earlier position in the manuscript, ahead of the section “Catalytic route determination...”.

4. In Figure 3e and Extended Data Fig. 1, the calculation is based on 1,3H transfer mechanisms, will be possible to have two 1,2H transfers instead of one 1,3H transfer? In that case, the energy might be lower.

Response: We greatly appreciate your genuine and professional advice. As suggested by you and the first reviewer, and combined our isotopic labeling results, we corrected our pathway ii (Extended Data Fig. 5) and recalculated it. The results were shown in Figure 4e and Figure S19.

5. The structures of MicA, Bnd4, AlbS have been analysed in extended Data Fig. 4. With these structures, some possible mechanisms may be proposed on the 1,10 or 1,14 cyclization of GGPP as proposed in Figure 3a.

Response: Thank you for your kind suggestions. The structural analysis of MicA, Bnd4, and AlbS is presented in Extended data Fig. 7 for the purpose of elucidating the rationale of the predicted MicA protein structure. As suggested, the cyclization mechanisms of MicA, Bnd4, and AlbS have been depicted in Figure S16. However, the lack of crystal structures for MicA, Bnd4, and AlbS hinders further mechanistic exploration at present. Comprehensive understanding will necessitate subsequent crystallographic studies.

6. Although compound **1** is definitely much more rigid than others, based on the DFT calculation, compound **1** still has small degree of flexibility. As such, is it accurate to define compound **1** as non-flexible? If not, the title may also need to be revised.

Response: We deeply appreciate your insightful feedback. Leveraging DFT calculations, we determined that compound **1** exhibits a minimal degree of flexibility (0.7%). Additionally, metadynamics simulations revealed $\Delta G \approx 3$ kcal/mol between two main conformers, with an energy barrier around 7.5 kcal/mol (Figure S13). These results align with our variable temperature nuclear magnetic resonance (NMR) spectroscopy findings: as the temperature reaches or exceeds 328K, the NMR spectrum broadens, indicating rapid conformational changes at high temperature (Figures S14 and 15). In light of this, we have characterized compound **1** as "nearly non-flexible" or

"almost inflexible" and have accordingly made the recommended adjustments in our manuscript.

REVIEWER COMMENTS

Reviewer #1 (Remarks to the Author):

The authors have adequately addressed the reviewers' comments, and I believe the manuscript is now suitable for publication with respect to computational chemistry.

The authors have done the re-calculations at the level of theory I suggested, and the results appear reliable. Furthermore, the IRC plots are provided in the supporting information, which confirms the validity of the findings.

The authors have also provided satisfactory responses to the comments raised by the other reviewers.

Reviewer #2 (Remarks to the Author):

The authors have addressed most of the comments from the referees. However, there are still problems especially in the DFT calculation and the explanation on how the trans-configuration of C1 and C10 with a 2Z double bond contributes to the reduced-flexibility of compound 1, as shown blow:

(DFT calculation):

Authors revised the "pathway ii". However, as the authors say, the pathway ii contains an obviously unfavorable step (1,2 H-shift in K+ to L+; Extended Data Fig. 5), in which the hydrogen atom in the lower-face of the molecule moves to the upper-face. I consider that there are more plausible pathways which do not contain the obviously unfavorable step. Even though the current "pathway ii" can be retained in the manuscript, the authors should also calculate other plausible pathways (Pathways A and B) as follows:

(Pathway A)

i) After G+ is formed, 1,2 H-shift occurs (the hydrogen atom on C-1 moves to C-10), to form the diastereomer of H+, whose stereochemistry at C-10 position is different from that of H+.
ii) the C-C bond between C-1 and C-14 is formed. iii) 1,2 H-shift occurs (the hydrogen atom on C14 moves to C15). iv) 1,2 H-shift occurs (the hydrogen atom on C1 move to C14). v) 1,2 H-shift occurs (the hydrogen atom on C-10, which is in the "upper-face" of the molecule,

moves to C-1). vi) 1,2 H-shift occurs (the hydrogen atom on C-11 moves to C-10) to form E+.

(Pathway B)

i) After F+ is formed, 1,3 H-shift occurs (the hydrogen atom on C-1 moves to C-11), to form the diastereomer of H+, whose stereochemistry at C-11 position is different from that of H+.
ii) the C-C bond between C-1 and C-14 is formed. iii) 1,2 H-shift occurs (the hydrogen atom on C14 moves to C15). iv) 1,2 H-shift occurs (the hydrogen atom on C1 move to C14). v) 1,3 H-shift occurs (the hydrogen atom on C-11 moves to C-1) to form E+.

Without the calculation of all possible pathways, the author cannot claim that the “pathway i” is the preferred one.

In addition, the author should calculate the reactions from A+ to B+ and from A+ to F+.

(the explanation on how the trans-configuration of C1 and C10 with a ZZ double bond contributes to the reduced-flexibility of compound 1):

The authors say that “We further hypothesized that the Z configuration of $\Delta_{2,3}$ limits the rotations of C1–C2 and C3–C4 bonds. The configurations of C1 and C10 might impact the rotation of the C1–C10 bond, which in turn influences the rotation of adjacent bonds. This cascade of effects leads to conformational changes, notably affecting the rotation of the C7–C19 bond.” This explanation should be improved. The authors should explain why the Z configuration of $\Delta_{2,3}$ limits the rotations of C1–C2 and C3–C4 bonds and why the configurations of C1 and C10 might impact the rotation of the C1–C10 bond. In addition, the authors also say that “the trans configuration of C1 and C10 could potentially offer enhanced thermodynamic stability”. However, the grounds for thinking so are not shown. The detailed explanation should be added.

(minor comment):

- Extended Data Fig. 5: The intermediates F+ and L+ should be included in the red box showing the pathway (ii). In addition, the intermediate E+ should be excluded from blue box showing pathway (i), because the intermediate E+ is formed from both D+ and L+.

Reviewer #3 (Remarks to the Author):

The authors have addressed all my concerns well. It is ready for acceptance except a minor suggestion. Fig 3b and extended data Fig 2 are quite similar, which makes one redundant, may consider deleting one.

Point-by-point Response

We express our gratitude to the reviewers for their thorough review and evaluation of our manuscript. We greatly value the constructive criticism and feedback provided. In the following sections, we address each comment in detail, point by point.

Reviewer #1

The authors have adequately addressed the reviewers' comments, and I believe the manuscript is now suitable for publication with respect to computational chemistry.

The authors have done the re-calculations at the level of theory I suggested, and the results appear reliable. Furthermore, the IRC plots are provided in the supporting information, which confirms the validity of the findings.

The authors have also provided satisfactory responses to the comments raised by the other reviewers.

Response: We would like to extend our sincerest gratitude for your meticulous review. Thank you for your support and very positive comment.

Reviewer #2:

1. The authors have addressed most of the comments from the referees. However, there are still problems especially in the DFT calculation and the explanation on how the *trans*-configuration of C1 and C10 with a 2Z double bond contributes to the reduced-flexibility of compound **1**, as shown below.

Response: We sincerely appreciate your support and recognition. In compliance with your guidance, we have thoroughly calculated other plausible pathways and elucidated the pertinent issues; please refer to the designated sections for more detailed insights.

2. DFT calculation:

Authors revised the "pathway ii". However, as the authors say, the pathway ii contains an obviously unfavorable step (1,2 H-shift in K⁺ to L⁺; Extended Data Fig. 5), in which the hydrogen atom in the lower-face of the molecule moves to the upper-face. I consider that there are more plausible pathways which do not contain the obviously unfavorable step. Even though the current "pathway ii" can be retained in the manuscript, the

authors should also calculate other plausible pathways (Pathways A and B) as follows:

(Pathway A)

i) After G^+ is formed, 1,2 H-shift occurs (the hydrogen atom on C-1 moves to C-10), to form the diastereomer of H^+ , whose stereochemistry at C-10 position is different from that of H^+ . ii) the C-C bond between C-1 and C-14 is formed. iii) 1,2 H-shift occurs (the hydrogen atom on C14 moves to C15). iv) 1,2 H-shift occurs (the hydrogen atom on C1 move to C14). v) 1,2 H-shift occurs (the hydrogen atom on C-10, which is in the “upper-face” of the molecule, moves to C-1). vi) 1,2 H-shift occurs (the hydrogen atom on C-11 moves to C-10) to form E^+ .

(Pathway B)

i) After F^+ is formed, 1,3 H-shift occurs (the hydrogen atom on C-1 moves to C-11), to form the diastereomer of H^+ , whose stereochemistry at C-11 position is different from that of H^+ . ii) the C-C bond between C-1 and C-14 is formed. iii) 1,2 H-shift occurs (the hydrogen atom on C14 moves to C15). iv) 1,2 H-shift occurs (the hydrogen atom on C1 move to C14). v) 1,3 H-shift occurs (the hydrogen atom on C-11 moves to C-1) to form E^+ .

Without the calculation of all possible pathways, the author cannot claim that the “pathway i” is the preferred one.

In addition, the author should calculate the reactions from A^+ to B^+ and from A^+ to F^+ .

Response: Thank you for your professional suggestion. We have carefully considered your suggestion and performed the calculations for pathways A and B (now renamed pathways 2A and 2B) as you have outlined. The calculation also included the reactions from A^+ to B^+ and from A^+ to F^+ . The results have been updated in Fig. 4e, Extended Data Fig. 4, Figs. S21–26, Tables S21 and S22. Through comparison, pathway 1 was determined as the preferred pathway. We have included detailed descriptions of the comparison process in the main text as shown below:

“In pathway 1, a sequence unfolds that is thermodynamically favorable, with the isopropyl cation B^+ being 6.93 kcal/mol less energetic than A^+ , the cembranyl cation C^+ being 4.05 kcal/mol lower than B^+ , and the following cembranyl intermediate D^+ being further reduced by 8.84 kcal/mol in energy. A portion of the cembranyl cation

undergoes deprotonation to form the cembrane structure (compound **2**), while the majority advances towards a further 1,10 cyclization to generate a eunicellane scaffold. However, the energy barrier for this 1,10 cyclization is 21.03 kcal/mol. It is hypothesized that the protein may enable a specific conformational change after the cembranyl cation's formation, creating a crucial intermediate conformation that aligns the C1 and C10–C11 double bond closely, thus favoring the reaction sequence towards 1,10 cyclization. However, pathway 2 presents a thermodynamically less favorable process. In pathway 2A, a 1,14-cyclization occurs following hydride shifts, with an energy barrier of 23.43 kcal/mol. Q⁺ is predisposed to undergo deprotonation over the subsequent reaction. Pathway 2B involves a final step of 1,3-hydride shift, presenting an energy barrier of 29.94 kcal/mol. In pathway 2C, K⁺ undergoes a 1,2-hydride shift to form L⁺, facing a substantial energy barrier of 62.18 kcal/mol. This high energy barrier is likely due to a configurational flip that takes place during the process. Collectively, pathway 1 is identified as the energetically preferred pathway.”

2. The explanation on how the *trans*-configuration of C1 and C10 with a 2Z double bond contributes to the reduced-flexibility of compound **1**:

The authors say that “We further hypothesized that the Z configuration of $\Delta^{2,3}$ limits the rotations of C1–C2 and C3–C4 bonds. The configurations of C1 and C10 might impact the rotation of the C1–C10 bond, which in turn influences the rotation of adjacent bonds. This cascade of effects leads to conformational changes, notably affecting the rotation of the C7–C19 bond.” This explanation should be improved. The authors should explain why the Z configuration of $\Delta^{2,3}$ limits the rotations of C1–C2 and C3–C4 bonds and why the configurations of C1 and C10 might impact the rotation of the C1–C10 bond. In addition, the authors also say that “the *trans* configuration of C1 and C10 could potentially offer enhanced thermodynamic stability”. However, the grounds for thinking so are not shown. The detailed explanation should be added.

Response: Thank you for your valuable and professional comments. In our current report, we proposed a mechanism that specifically combined stereochemistry influences the flexibility of eunicellane scaffolds. However, as mentioned by Prof.

Andrei K. Yudin (*Chem. Rev.*, 2019, 119, 17, 9724–9752), the conformational analysis of macrocycles is a complex and challenging problem. Thanks to your professional and insightful suggestions, we have gone slightly further in seeking more explanations. We have included more detailed explanations based on your insightful suggestion, as below:

(1) Improved explanations based on your suggestions:

Prof. Andrei K. Yudin has previously proposed that alteration in backbone stereochemistry can lead to coupled bond rotations that can result in conformational reorganizations among macrocycles (*Chem. Rev.*, 2019, 119, 9724–9752). Based on that proposal, we hypothesized that the configurations of C1, C2/C3, and C10 applied similar mechanisms for adjusting the flexibility of the eunicellane macrocycle. In addition, the authors of “*Chem. Rev.*, 2019, 119, 9724–9752” have provided explanations of the mechanism concerning the dihedral angles of macrocycle-containing compounds. Accordingly, we conducted a thorough conformational analysis of all related eunicellanes and discovered that compounds **1** and *1R,10R-1* possess dihedral angles χ_1 (C9-C10-C1-C2) that are greater than 130° , whereas the other compounds exhibit less than 90° (Fig.s S16, S17, and Table S20). The *trans*-configuration of C1 and C10 with a *2Z* double bond might collectively influence the rotation of the C1–C2, C1–C10, and C9–C10 bonds, resulting in augmenting the dihedral angle, thereby rendering the rotations of remaining bonds, ultimately diminishing the structural flexibility. Our observation and hypothesized mechanism align with several previous reports. For example, Qingzhou Zhang et al. (*Nat. Sci. Rep.*, 2016, 6, 38573) previously reported that inducing α -helicity through side-chain cross-linking is a strategy that has been pursued to improve peptide conformational rigidity and bioavailability. The χ_1 (N-C-C-S) dihedral angle change from -66° (sulfoxide) to -167° (*S*-stereoisomer) renders the linker too short to maintain the α -helix.

(2) Reason for mentioning the effect of *trans* configuration of C1 and C10:

The reason for proposing that “the *trans* configuration of C1 and C10 could potentially

offer enhanced thermodynamic stability” is based on the intrinsic chemical characteristic of bicyclic *cis*- and *trans*-decalins. The decalins have two major configurational isomers, the *cis*-decalin and the *trans*-decalin (*Stereochemistry*, 2021, 273-375). In *cis*-decalin, the ring fusion is through axial-equatorial bonds of both cyclohexane rings, which can undergo ring flips. In *trans*-decalin, the ring fusion of one cyclohexane unit to the other unit is through equatorial-equatorial bonds, which have no ring inversion. The bonds between C-9 and C-10 in *trans*-decalin are rigidly fixed, preventing free rotations. The *cis*-decalin has a range between 8.8 and 11.4 kJ/mol higher enthalpy than the *trans*-decalin, so the *trans*-decalin is more stable than *cis*-decalin. To avoid confusion and misunderstanding, we have omitted relevant descriptions as it is repeated mention of the effect of the *trans* configuration of C1 and C10.

Finally, as for resolving the challenging research problem concerning the conformational analysis of macrocycles, substantial studies need to be performed by more experts in the fields of chemical calculations and crystallography in the future. Our discovery of this cryptic enzyme that can form the nearly non-flexible eunicellane scaffold indeed facilitates relevant studies. Moreover, understanding the principles that enhance or reduce of macrocycle might also facilitate the drug design in terms of conformational control of the eunicellane-type macrocycle.

4. Extended Data Fig. 5: The intermediates F^+ and L^+ should be included in the red box showing the pathway (ii). In addition, the intermediate E^+ should be excluded from blue box showing pathway (i), because the intermediate E^+ is formed from both D^+ and L^+ .

Response: Thank you for your valuable feedback. We have modified Extended Data Fig. 5 (now Extended Data Fig. 4). To avoid confusion, we opted to remove the boxes and employ colored arrows and description to provide detailed explanations for each pathway.

Reviewer #3:

The authors have addressed all my concerns well. It is ready for acceptance except a minor suggestion. Fig 3b and Extended data Fig 2 are quite similar, which makes one redundant, may consider deleting one.

Response: Thank you for your thorough review and for your positive feedback. We have carefully considered your suggestion and removed the Extended Data Fig. 2 in the revised version.

REVIEWERS' COMMENTS

Reviewer #2 (Remarks to the Author):

The authors have addressed my previous comments well. There is one minor suggestion shown below.

-lines 250 – 251: The explanation of the pathway 2 is incomplete. Two or three 1,2-hydride shifts after the 1,14-ring closure are missing. Please add the description about it.

Point-by-point Response

We express our gratitude to the reviewer #2 for the thorough review and evaluation of our manuscript. We greatly value the constructive criticism and feedback provided. In the following sections, we address the comment in detail, point by point.

Reviewer #2

The authors have addressed my previous comments well. There is one minor suggestion shown below.

-lines 250 – 251: The explanation of the pathway 2 is incomplete. Two or three 1,2-hydride shifts after the 1,14-ring closure are missing. Please add the description about it.

Response: We sincerely appreciate your support and suggestion. According to your suggestion, we have added a comprehensive supplementary description of pathway 2. The specific description is as follows (in lines 242–247): *“Pathway 2A and 2C initiate with a 1,10-cyclization, proceeding through two 1,2-hydride shifts, a 1,14-ring closure, and four additional 1,2-hydride shifts, culminating in deprotonation. Pathway 2B, on the other hand, initiates similarly with a 1,10-cyclization, but diverges with a 1,3-hydride shift, followed by a 1,14-ring closure, two 1,2-hydride shifts, another 1,3-hydride shift, and concludes with deprotonation”*.